# DiffVAS: Diffusion-Guided Visual Active Search in Partially Observable Environments

## Abstract

Visual active search (VAS) has been introduced as a modeling framework that leverages visual cues to direct aerial (e.g., UAV-based) exploration and pinpoint areas of interest within extensive geospatial regions. Potential applications of VAS include detecting hotspots for rare wildlife poaching, aiding in search-and-rescue missions, and uncovering illegal trafficking of weapons, among other uses. Previous VAS approaches assume that the entire search space is known upfront, which is often unrealistic due to constraints such as a restricted field of view and high acquisition costs, and they typically learn policies tailored to specific target objects, which limits their ability to search for multiple target categories simultaneously. In this work, we propose *DiffVAS*, a target-conditioned policy that searches for diverse objects simultaneously according to task requirements in partially observable environments, which advances the deployment of visual active search policies in real-world applications. DiffVAS uses a diffusion model to reconstruct the entire geospatial area from sequentially observed partial glimpses, which enables a target-conditioned reinforcement learning-based planning module to effectively reason and guide subsequent search steps. Our extensive experiments demonstrate that DiffVAS excels in searching diverse objects in partially observable environments, significantly surpassing state-of-the-art methods across datasets.

## 1 Introduction

Consider a scenario where a search-and-rescue mission is underway, and rescue personnel needs to scan across hundreds of potential regions from a helicopter to locate a missing person. A crucial strategy in such operations involves using UAVs to capture aerial imagery that can help identify a target of interest (e.g., the missing person). However, constraints like a limited field of view, high acquisition costs, time constraints, and restricted bandwidth between the sensor and the processing unit can make the search extremely challenging, demanding strategic decisions on where to query next based on the observations gathered so far. A similar challenge arises in other scenarios, such as locating a specific vehicle in an abduction case – however, note that *the target may differ, but the underlying problem structure remains the same*. In fact, many other scenarios share this general structure, such as anti-poaching enforcement (Fang et al., 2015), pinpointing landmarks, identifying drug or human trafficking sites, and more (Fang et al., 2016; Bondi et al., 2018).

In this work, we derive and formalize a general task setup that encompasses these types of scenarios, and that allows for controllable and reproducible model development and experimentation. We refer to our proposed task setup as *Target-Conditioned Visual Active Search in Partially Observable environments (TC-POVAS)*, the details of which are given in Sec. 2. The setup of TC-POVAS is as follows: Given a target category (or multiple target categories, depending on the task requirement), one should leverage a series of partially observed glimpses – which are sequentially queried during active exploration – to locate as many target objects as possible. Note that the number of allowed queries is limited in TC-POVAS, to reflect factors such as time or resource constraints.

TC-POVAS builds on the visual active search (VAS) framework in which one aims to find a target object using visual cues through sequential exploration (Sarkar et al., 2023; 2024a). Past work on visual active search (VAS) has assumed access to a complete description of the search space (typically an image that spans the whole area) for making decisions. However, in many real-world situations, e.g. search-and-rescue operations, an entire image of the search space may not be available upfront.

For example, an autonomous drone on a rescue mission might only be able to capture partial glimpses through a series of narrow observations due to constraints like a confined viewing range and high data collection costs. In these scenarios, the agent has to make decisions with incomplete information, and thus models trained assuming access to complete images will struggle.

The challenge is twofold: **(i)** the agent must query the most informative patch from a partially observed scene to maximize information gain about the search space, and **(ii)** it must simultaneously ensure that this patch helps achieve the goal of locating regions containing the target objects. One might question why an agent cannot simply learn to choose patches that reveal target regions directly, without the need for acquiring knowledge about the underlying scene. The challenge arises because reasoning in an unknown partially observable environment is inherently difficult. Thus, an agent must strike a balance between *exploration* – identifying patches that reveal the most information about the search space – and *exploitation* – focusing on areas likely to contain the target object based on updated knowledge about the environment. An optimal agent must master this delicate balance to be effective. Additionally, previous VAS policies (Sarkar et al., 2023; 2024a) are designed to search for specific target objects and cannot handle multiple categories simultaneously, which limits their adaptability to specific task preferences.

To address these challenges, and to effectively tackle the TC-POVAS task setting, we propose *DiffVAS*, a novel framework that consists of two key modules: **(1)** a diffusion-based *conditional generative module (CGM)* and **(2)** a *target-conditioned planning module (TCPM)*. The goal of the CGM is to learn how to reconstruct the entire scene (search space) contingent on the partially observed glimpses gathered so far. To achieve this, we employ a neural network architecture that enables precise control over image generation by conditioning the diffusion-based generative model on the partially observed glimpses. Such a CGM attains fine control over image generation by integrating input conditions, like previously observed glimpses, directly into the model's intermediate layers, influencing the output at various stages of the diffusion process. This layered integration allows the model to align closely with the input conditions, ensuring that the generated image adheres to the desired structure while benefiting from the diffusion model's generative capabilities.

The objective of the TCPM is to decide which patch to query next by analyzing the scene generated by the CGM and the partially observed glimpses, with the aim of revealing as many target regions as possible within the query budget. To accomplish this, the TCPM must learn to simultaneously explore the environment efficiently to maximize information gathering (exploration) *and* select patches that reveal as many target regions as possible based on its acquired knowledge of the environment (exploitation). To this end, we develop an RL-based policy that learns to balance exploration and exploitation. To train the policy, we design a reward function that – besides encouraging target discovery – takes into account two key factors: *local uncertainty* and *global reconstruction quality*. Together, these factors measure how effectively the policy issues actions that contribute to gaining information about the environment. Additionally, we designed the TCPM to be target-conditioned, which enables it to search for different target categories according to task requirements and handle multiple categories simultaneously. We accomplish this by introducing an inference strategy that leverages target-conditioned probability distributions over grid cells for each target category, computed via TCPM, and learning target-aware state representation by leveraging cross-attention. Finally, we conduct extensive experiments to demonstrate the effectiveness of *DiffVAS*.

In summary, we make the following contributions:

- We introduce TC-POVAS, a novel task setup that addresses target-conditional (TC) visual active search (VAS) in partially observable (PO) environments, and which extends traditional VAS to become more closely aligned with practical scenarios.
- We propose *DiffVAS*, an agent that effectively tackles this task by reconstructing the whole search area as it explores and searches for targets. Unlike previous approaches, DiffVAS can search a diverse range of target objects and tackle multiple target categories simultaneously.
- We demonstrate the significance of each component within DiffVAS through a comprehensive series of quantitative and qualitative ablation analyses.
- Our extensive experimental evaluations using two publicly available satellite imagery datasets (xView and DOTA), across various unknown target settings, demonstrate that DiffVAS significantly outperforms all baseline approaches. The code and models will be made public.

## 2 TC-POVAS Task Setup

In this section, we describe the details of our proposed TC-POVAS task setup; see Fig. 1 for an overview. TC-POVAS is a search task in which one or multiple targets should be localized within a search area – represented here as an aerial image $x$ that is partitioned into $N$ grid cells, such that $x = (x^{(1)}, x^{(2)}, ..., x^{(N)})$ – within a given query budget $\mathcal{B}$ which here represents the number of movement actions. Each grid cell corresponds to a sub-image and represents the limited field of view of the agent (akin to a UAV hovering at a limited altitude), i.e., the agent can only observe the aerial content of a sub-image $x^{(i)}$ corresponding to the $i$th grid cell in which it is located at time step $t$. The agent's action space corresponds to all possible movements to other grid cells.

For each task configuration, the target object categories are predefined in natural language, such as "small car, boat", and represented as a set $\mathcal{Z}$. The objective is to uncover as many grid cells as possible that contain objects in $\mathcal{Z}$ by strategically exploring the grid cells within the budget constraint $\mathcal{B}$. To keep track of which grid cells $x^{(j)}$ contain targets, we label each grid cell $x^{(j)}$ with $y^{(j)}(\cdot \mid \mathcal{Z}) \in \{0, k\}$, where

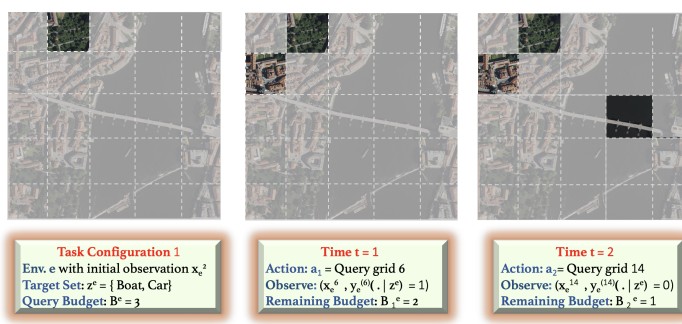

Figure 1: An overview of the TC-POVAS task setup.

$y^{(j)}(\cdot \mid \mathcal{Z}) = k$ if cell $j$ contains at least one instance each of $k$ different target object categories from set $\mathcal{Z}$, and 0 otherwise. The full label vector for the task is $y(\cdot \mid \mathcal{Z}) = (y^{(1)}(\cdot \mid \mathcal{Z}), y^{(2)}(\cdot \mid \mathcal{Z}), ..., y^{(N)}(\cdot \mid \mathcal{Z}))$. Naturally, at decision time we assume no direct knowledge of $y(\cdot \mid \mathcal{Z})$, but it is used to evaluate an agent's task performance at the end of an episode. Moreover, when an agent queries a grid cell $j$, it receives $x^{(j)}$ (the aerial image content of the $j$:th grid cell) and the corresponding ground truth label $y^{(j)}(\cdot \mid \mathcal{Z})$ for that cell.[1] An overview of the search task is provided in Fig. 1. Denoting a query performed in step $t$ as $q_t$ and $c(i, j)$ as the cost associated with querying grid cell $j$ starting from grid cell $i$, the overall task optimization objective is:

$$\max_{\{q_t\}} \sum_t y^{(q_t)}(\cdot \mid \mathcal{Z}) \text{ subject to } \sum_{t \geq 0} c(q_{t-1}, q_t) \leq \mathcal{B} \tag{1}$$

**Target-Conditioned Partially Observable Markov Decision Process (TC-POMDP).** With objective (1) in mind, we aim to learn a search policy that can efficiently explore a search area and discover target regions, and to achieve this through learning from similar pre-labeled search tasks, referred to as $\mathcal{D} = \{(x_i, y_i(\cdot \mid \mathcal{Z}))\}$, which consists of images $x_i$ paired with corresponding grid cell labels $y_i(\cdot \mid \mathcal{Z})$. Here, each $x_i$ is composed of $N$ elements $(x_i^{(1)}, x_i^{(2)}, \ldots, x_i^{(N)})$ which represent the grid cells in the image, and each $y_i(\cdot \mid \mathcal{Z})$ contains $N$ corresponding labels $y_i^{(1)}(\cdot \mid \mathcal{Z}), y_i^{(2)}(\cdot \mid \mathcal{Z}), \ldots, y_i^{(N)}(\cdot \mid \mathcal{Z})$. We model this problem as a TC-POMDP and consider a family of TC-POMDP environments $\mathcal{M}^e = \{(\mathcal{S}^e, \mathcal{A}, \mathcal{X}^e, \mathcal{T}^e, \mathcal{G}^e, \gamma) \mid e \in \epsilon\}$, where $e$ is the environment index. Each environment $\mathcal{M}^e$ comprises a state space $\mathcal{S}^e$, shared action space $\mathcal{A}$, observation space $\mathcal{X}^e \in \{(x_e^{(1)}, x_e^{(2)}, \ldots, x_e^{(N)})\}$, transition dynamics $\mathcal{T}^e$, target space $\mathcal{G}^e(\mathcal{Z}) \subset \mathcal{S}^e$ such that $\mathcal{G}^e(\mathcal{Z}) = \{x_e^{(g)} \in \mathcal{X}^e \mid y_e^{(g)}(\cdot \mid \mathcal{Z}) \neq 0 \text{ for } g \in \{1, 2, \ldots, N\}\}$, and discount factor $\gamma \in [0, 1]$. $\mathcal{T}^e$ involve updating the remaining budget $\mathcal{B}_{t+1}$ by subtracting the current query cost $c(q_{t-1}, q_t)$ and incorporating the latest query outcomes, i.e. $x_e^{(q_t)}, y_e^{(q_t)}(\cdot \mid \mathcal{Z})$, into the state at time $t + 1$. The observation $x^e \in \mathcal{X}^e$ is determined by state $s^e \in \mathcal{S}^e$ and the unknown environmental factor $b^e \in \mathcal{F}^e$, i.e. $x^e(s^e, b^e)$, where $\mathcal{F}^e$ encompasses variations (including seasonality, weather effects, etc) related to diverse geospatial regions. $x_e^{(q_t)}$ denote the observation associated with $q_t$ at step $t$, for domain $e$.

The primary objective in a TC-POMDP is to learn a history-aware target-conditioned policy $\pi(a_t \mid x_{h_t}^e, \mathcal{Z}, \mathcal{B}_t^e)$ – where $x_{h_t}^e = (x_e^{(q_1)}, \ldots, x_e^{(q_t)})$ combines all the previous observations up to

---

[1]It would also be possible to consider a setting where an aerial object detector is used to assess what objects are within a grid cell.

time $t$, $\mathcal{B}_t^e$ represents the remaining budget at time $t$ – that maximizes the discounted state density function $J(\pi)$ across all domains $e \in \epsilon$ as follows:

$$J(\pi) = \mathbb{E}_{e \sim \epsilon, \mathcal{B}_0^e \sim \mathcal{B}^e, \mathcal{Z} \sim \text{RandomSubset}(\mathcal{O}^e), \pi} \left[ (1-\gamma) \sum_{t=0}^{\infty} \gamma^t p_\pi^e(s_t \in \mathcal{G}^e(\mathcal{Z}) | \mathcal{Z}, \mathcal{B}_t^e) \right] \qquad (2)$$

Here $p_\pi^e(s_t \in \mathcal{G}^e(\mathcal{Z}) | \mathcal{Z}, \mathcal{B}_t^e)$ represents the probability of querying a grid cell containing at least one target at step $t$ within domain $e$ under the policy $\pi(.|x_{h_t}^e, \mathcal{Z}, \mathcal{B}_t^e)$, $\mathcal{O}^e$ denotes the set of object categories in domain $e$, and $e \sim \epsilon, \mathcal{B}_0^e \sim \mathcal{B}^e$ refer to uniform samples from each set. The total query budget allocated for a search task is denoted as $\mathcal{B}^e$. Throughout the training process, the agent is exposed to a set of training environments $\{e_i\}_{i=1}^N = \epsilon_{\text{train}} \subset \epsilon$, each identified by its environment index. To reduce clutter, we omit the notation $e$ for the rest of the paper. Next, we explore how we design and train a policy – which we call *DiffVAS* – to effectively maximize the objective outlined in (2).

## 3 DIFFVAS: A DIFFUSION-GUIDED APPROACH FOR TACKLING TC-POVAS

In this section we introduce *DiffVAS*, a diffusion-guided, reinforcement learning (RL)-based agent designed to address VAS in partially observable environments. DiffVAS is composed of two main modules: (1) a conditional generative module (CGM) and (2) a target-conditioned planning module (TCPM). Next, we detail each component of the proposed DiffVAS framework, starting with the training strategy for both modules to learn an efficient policy, followed by the inference procedure.

### 3.1 TRAINING

Our approach uses a two-phase training strategy: In the first phase, we train the CGM, and then we freeze its parameters while training the TCPM in the second phase. The purpose of the CGM is to synthesize the entire scene (i.e., the search space) from the partially observed glimpses collected so far, thereby assisting the TCPM in deciding the next query location. To achieve this, the conditional generative model leverages a diffusion-based adapter-style approach (Mou et al., 2024; Zhang et al., 2023).

Diffusion models are powerful generative models that allow for precise control over the attributes of the generated samples. While these diffusion models trained on large datasets have achieved success, there is often a need to introduce additional controls in downstream fine-tuning processes. In our case, the CGM finetunes the diffusion model by integrating information about previously observed glimpses $x_{h_t}$ while preserving the integrity of the pretrained diffusion model. This is done by freezing the parameters

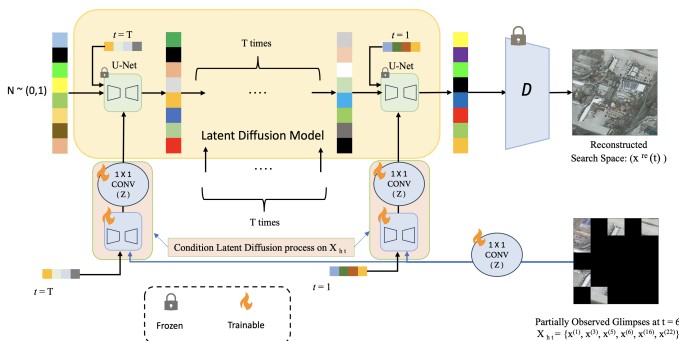

Figure 2: The conditional generative module within DiffVAS.

of a trained diffusion model and creating a trainable copy that takes an external conditioning vector $x_{h_t}$ as input (see Fig. 2). The trainable copy is connected to the frozen pre-trained diffusion model using zero convolution layers $Z(;)$, which are $1 \times 1$ convolution layers initialized with weights and biases set to zero, safeguarding the model against any harmful noise in the early stages of training, as outlined in (Zhang et al., 2023). This design strategy thus retains the capabilities of the large-scale pre-trained diffusion model while allowing the trainable copy to adapt to new conditions.

To train the parameters of the CGM, we randomly sample an image $x_0$ corresponding with an entire search space, and progressively add noise to create a noisy image $x_k$, where $k$ indicates the number of noise additions. Conditioned on partially observed glimpses $x_{h_t}$, CGM trains a network $\epsilon_\theta$ to predict the noise added to $x_k$ using the following equation:

$$\mathcal{L}_{CGM} = \mathbb{E}_{x_0, k, x_{h_t}, \epsilon \sim \mathcal{N}(0,1)} \left[ \| \epsilon - \epsilon_\theta(x_k, k, x_{h_t}) \|_2^2 \right] \qquad (3)$$

$\mathcal{L}_{CGM}$ represents the overall learning objective of the CGM. Note that $x_{h_t}$ is obtained by randomly selecting a history length $h_t \in \{1, \ldots, N-1\}$, then choosing $h_t$ random patches while masking the

rest of $x_0$. An overview of the CGM is presented in Fig. 2, with detailed architecture and training hyperparameters provided in the appendix. Next, we discuss the training procedure for the TCPM.

**TCPM training.** The role of the TCPM is to determine the next query location based on $x_{h_t}$, $\mathcal{B}^t$, and the target category $\mathcal{Z}$. The planning module must *explore* – seeking patches that provide the most insight into the search space – while also *exploiting* known information, focusing on areas with a high likelihood of containing the target. To this end, we develop an actor-critic style PPO algorithm (Schulman et al., 2017) for learning a policy that balances exploration and exploitation, which is essential for solving this task. Since decision-making in an unknown environment is challenging, we leverage the trained CGM to reconstruct the entire search space $x_{\text{re}}(t)$ from partially observed glimpses $x_{h_t}$. This reconstructed information aids the planning module in making more informed decisions about the next query location. As illustrated in Fig. 3, the latent representation $l_{\text{re}}(t)$ of $x_{\text{re}}(t)$ is extracted from the encoder at the final step of the reverse diffusion process of the pre-trained CGM (i.e., $x_{\text{re}}(t) = D(l_{\text{re}}(t) = \text{CGM}(x_{h_t}))$. We use the encoder $e^{\text{CGM}}(\cdot)$ of the CGM as a feature extractor to derive the latent representation $l_h(t)$ of $x_{h_t}$, i.e. $l_h(t) = e^{\text{CGM}}(x_{h_t})$. We merge $l_{\text{re}}(t)$ and $l_h(t)$ channel-wise, forming the combined representation $l_{\text{img}}(t)$. The key reason for incorporating $l_h(t)$ into the state space is that early in the search, the reconstruction $x_{\text{re}}(t)$ of the search space may be unreliable, making it imprudent to base decisions solely on $l_{\text{re}}(t)$.

As we want to learn a policy capable of searching for diverse target objects, we condition it on the target object $z$. Here, $z$ is an element of the set of target object categories (i.e., $z \in \mathcal{Z}$; see Sec. 3.2 for how the multi-target setting is handled). The target object embedding $l_z$ is obtained via the CLIP (Radford et al., 2021) text encoder (i.e., $l_z = f^{\text{CLIP}}(z)$). A learnable cross-attention layer is then applied between $l_z$ and $l_{\text{img}}(t)$, which allows us to obtain a representation of the search space that is target-aware, denoted as $l^z_{\text{img}}(t)$. At time $t$, the planning module's input state comprises $l_{\text{img}}(t)$, $l_z$, the remaining budget $\mathcal{B}^t$, and an observation vector $o^t(\cdot \mid z)$ that encodes previous search query outcomes. Each element of $o^t(\cdot \mid z)$ corresponds to a grid cell index, where $o^t_{(j)}(\cdot \mid z) = 2y^{(j)}(\cdot \mid z) - 1$ if the $j$:th grid cell has been explored, and $o^t_{(j)}(\cdot \mid z) = 0$ otherwise. The primary reason for incorporating $\mathcal{B}^t$ and $o^t(\cdot \mid z)$ into the state space is to ensure that the planning module makes decisions with full awareness of both remaining budget and previous query outcomes.

Let us denote the state at time $t$ as $s_t = [l_{\text{img}}(t), l_z, o^t(\cdot \mid z), \mathcal{B}^t]$. Training TCPM is done using PPO Schulman et al. (2017) and involves learning both an *actor* (policy network, parameterized by $\zeta$) $\pi_\zeta : s_t \to p(\mathcal{A})$ and a *critic* (value function, parameterized by $\eta$) $V_\eta : s_t \to \mathbb{R}$ that approximates the true value $V^{\text{true}}(s_t) = \mathbb{E}_{a \sim \pi_\zeta(.|l_{\text{img}}(t), l_z, o^t(\cdot|z), \mathcal{B}^t)}[R(s_t, a_t, z) + \gamma V(\mathcal{T}(s_t, a_t))]$. We optimize both the actor and critic networks using the following loss function:

$$\mathcal{L}^{\text{planner}}_t(\zeta, \eta) = \mathbb{E}_t \left[ -\mathcal{L}^{\text{clip}}(\zeta) + \alpha \mathcal{L}^{\text{crit}}(\eta) - \beta \mathcal{H}[\pi_\zeta(.|l_{\text{img}}(t), l_z, o^t(\cdot \mid z), \mathcal{B}^t)] \right] \quad (4)$$

Here $\alpha$ and $\beta$ are hyperparameters, and $\mathcal{H}$ denotes entropy, so minimizing the final term of (4) encourages the actor to exhibit more exploratory behavior. The $\mathcal{L}^{\text{crit}}$ loss is used specifically to optimize the parameters of the critic network and is defined as a squared-error loss, i.e. $\mathcal{L}^{\text{crit}} = (V_\eta(l_{\text{img}}(t), l_z, o^t(\cdot \mid z), \mathcal{B}^t) - V^{\text{true}}(s_t))^2$. The clipped surrogate objective $\mathcal{L}^{\text{clip}}$ is employed to optimize the parameters of the actor-network while constraining the change to a small value $\epsilon$ relative to the old actor policy $\pi^{\text{old}}$ and is defined as:

$$\mathcal{L}^{\text{clip}}(\zeta) = \min \left\{ \frac{\pi_\zeta(.|l_{\text{img}}(t), l_z, o^t(\cdot \mid z), \mathcal{B}^t)}{\pi^{\text{old}}(.|l_{\text{img}}(t), l_z, o^t(\cdot \mid z), \mathcal{B}^t)} A^t, \text{clip}\left(1 - \epsilon, 1 + \epsilon, \frac{\pi_\zeta(.|l_{\text{img}}(t), l_z, o^t(\cdot \mid z), \mathcal{B}^t)}{\pi^{\text{old}}(.|l_{\text{img}}(t), l_z, o^t(\cdot \mid z, \mathcal{B}^t)})A^t \right) \right\}$$
$$A^t = r_t + \gamma r_{t+1} + \ldots + \gamma^{T-t+1} r_{T-1} - V_\eta(l_{\text{img}}(t), l_z, o^t(\cdot \mid z), \mathcal{B}^t) \quad (5)$$

After every fixed update step, we copy the parameters of the current policy network $\pi_\zeta$ onto the old policy network $\pi^{\text{old}}$ to enhance training stability. All hyperparameter details for training the actor and critic network are in Appendix A5. Our proposed DiffVAS framework is illustrated in Fig. 3. Next, we introduce a novel reward function $\mathcal{R}$ designed to guide the planning module in mastering an efficient search strategy in partially observed scenes.

**Reward structure.** The reward $\mathcal{R}$ consists of three components: (i) *local uncertainty* reward $\mathcal{R}^{\text{LU}}$, (ii) *global reconstruction* reward $\mathcal{R}^{\text{GR}}$, and (iii) *active search* reward $\mathcal{R}^{\text{AS}}$. The $\mathcal{R}^{\text{LU}}$ and $\mathcal{R}^{\text{GR}}$ rewards are designed to assess how efficiently the planning module's choice of movement enhances information-gathering about the environment (*exploration*), whereas $\mathcal{R}^{\text{AS}}$ assesses how well the

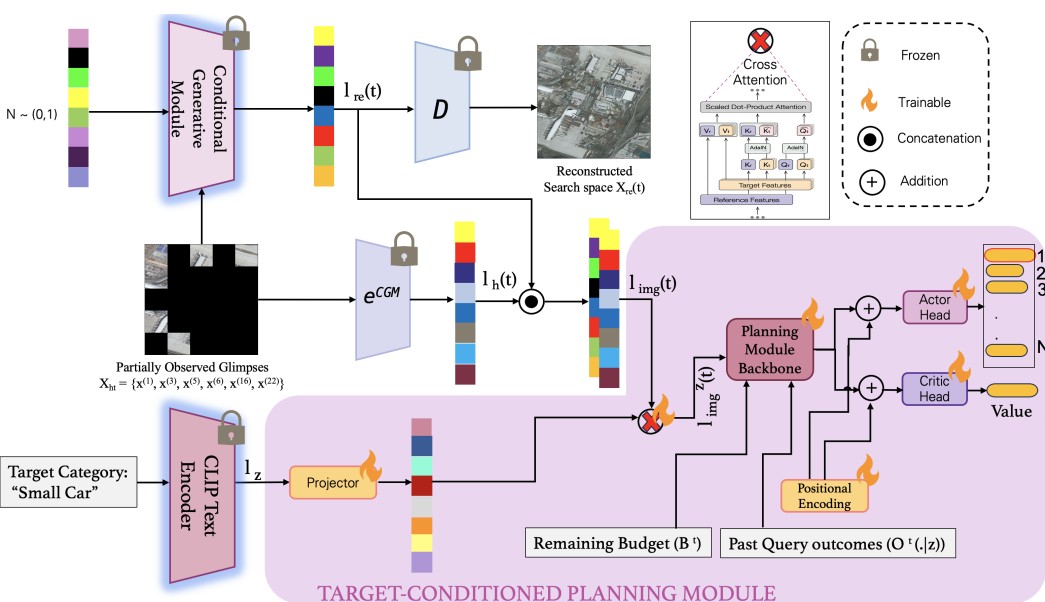

Figure 3: DiffVAS framework for visual active search in partially observable environments.

policy is discovering target regions (*exploitation*). We define $\mathcal{R}^{\text{LU}}$ as follows:

$$\mathcal{R}^{\text{LU}} = \text{sgn}\left[\left\{\text{SSIM}\left(x_{\text{true}}^{(a_{\text{ran}})}, D\left(\text{CGM}(x_{h_{t-1}})\right)^{(a_{\text{ran}})}\right)\right\} - \left\{\text{SSIM}\left(x_{\text{true}}^{(a_t)}, D\left(\text{CGM}(x_{h_{t-1}})\right)^{(a_t)}\right)\right\}\right] \quad (6)$$

where, the structural similarity index (Wang & Bovik, 2002) $\text{SSIM}(a, b)$ is used to measure the similarity between two images $a$ and $b$; $a_{\text{ran}}$ represents a randomly selected grid cell at time $t$; $x_{\text{true}}^{(a_{\text{ran}})}$ and $x_{\text{true}}^{(a_t)}$ refer to the $a_{\text{ran}}$:th and $a_t$:th grid cells of the ground truth image, respectively. According to (6), the agent receives a positive reward when the ground truth and reconstructed patches are more *dissimilar* (according to the SSIM score) for the queried grid cell than for a randomly selected grid cell index (i.e., $a_{\text{ran}}$). Thus, (6) gives a positive reward when the agent queries a patch that it is uncertain of, encouraging the discovery of novel (and uncertain) parts of the overall search area.

As for the global reconstruction reward, it is defined similarly as follows:

$$\mathcal{R}^{\text{GR}} = \text{sgn}\left[\left\{\text{SSIM}\left(x_{\text{true}}, D\left(\text{CGM}(x_{h_t})\right)\right)\right\} - \left\{\text{SSIM}\left(x_{\text{true}}, D\left(\text{CGM}(x_{h_t^{\text{ran}}})\right)\right)\right\}\right] \quad (7)$$

where $x_{h_t^{\text{ran}}}$ is identical to $x_{h_t}$, except the action $a_t$ at time $t$ is replaced with a random action $a_{\text{ran}}$. $\mathcal{R}^{\text{GR}}$ rewards the agent if querying the grid cell ($a_t$) results in a *better* reconstruction of the entire search space by the CGM module compared to querying a random grid cell such as $a_{\text{ran}}$ – thus note that this reward term is in some sense "inverse" relative to (6). In the early stages of the search, the search space reconstruction by CGM is poor (see an example in Fig. 4) regardless of the queried grid cell, making the $\mathcal{R}^{\text{GR}}$ reward signal weak. Therefore, relying solely on $\mathcal{R}^{\text{GR}}$ is not effective for distinguishing between good and bad grid cell selections. In this scenario, $\mathcal{R}^{\text{LU}}$ offers a sharper distinction, as it is based on evaluating a single grid cell.

To ensure the agent's queried grid cell also contributes to identifying regions with the target object, we design an active search reward function $\mathcal{R}^{\text{AS}}$ defined as $\mathcal{R}^{\text{AS}} = y^{(a_t)}(\cdot \mid z)$. Thus, the agent receives a positive reward for querying an unexplored cell containing a target; otherwise, $\mathcal{R}^{\text{AS}} = -5$, which penalizes the agent heavily for querying the grid cell more than once. Finally, we train the agent using the following reward function:

$$\mathcal{R}(s_t, a_t, z) = \mathcal{R}^{\text{LU}} + \mathcal{R}^{\text{GR}} + \mathcal{R}^{\text{AS}} \quad (8)$$

Next, we discuss the inference procedure of our proposed DiffVAS framework.

## 3.2 INFERENCE

In this section we outline the approach for searching one or multiple target categories simultaneously, based on task requirements, using the trained DiffVAS agent. Denote the set of target objects to be

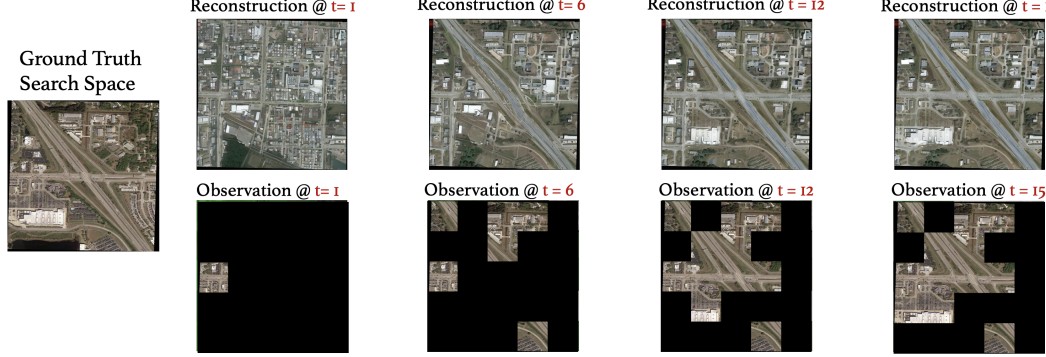

Figure 4: CGM's reconstruction from partially observed glimpses at various search stages.

searched as $\mathcal{Z} = \{z_1, \ldots, z_k\}$. At each search step, we compute the probability of querying each grid cell, conditioned on the $i$'th element of set $\mathcal{Z}$, using the trained DiffVAS, representing the resulting distribution over grid cell as $\pi_\zeta(.|l_{\text{img}}(t), l_{z_i}, o^t(\cdot \mid z_i), \mathcal{B}^t)$. We independently compute such conditional probability distribution for each element in set $\mathcal{Z}$ and select a grid cell to query based on the joint probability distribution, defined as follows:

$$\pi_\zeta(\cdot \mid l_{\text{img}}(t), l_\mathcal{Z}, o^t(\cdot \mid \mathcal{Z}), \mathcal{B}^t) = \prod_{c=1}^k p_c \quad \text{where} \quad p_c = \pi_\zeta(\cdot \mid l_{\text{img}}(t), l_{z_\mathbf{c}}, o^t(\cdot \mid z_\mathbf{c}), \mathcal{B}^t) \quad (9)$$

---

**Algorithm 1** Inference procedure of DIFFVAS

---

**Require:** Task instance with initial observation$(x^{(\text{init})}, y^{(\text{init})})$; set of target objects $\mathcal{Z} = \{z_1, \ldots, z_k\}$; budget $\mathcal{B}$; trained CGM; encoder $e^{\text{CGM}}$ of CGM; CLIP text encoder $f^{\text{CLIP}}$; trained TCPM parameters $(\zeta, \eta)$.
1: **Initialize** $o^0(\cdot \mid z_c) = [0...0]$ for each $c \in \{1, \ldots, k\}$; $\mathcal{B}^0 = \mathcal{B}$; $x_{h_0} = \{x^{(\text{init})}\}$; step $t = 0$; $R^{\text{task}} = 0$
2: **while** $\mathcal{B}^t > 0$ **do**
3:     $l_{\text{img}}(t) = \text{CGM}(x_{h_t}) \oplus e^{\text{CGM}}(x_{h_t})$, where $\oplus$ represents channel-wise concatenation operation.
4:     **for** $c = 1$ to $k$ **do**
5:         Compute $l_{z_c} = f^{\text{CLIP}}(z_c)$, and $p_c = \pi_\zeta(\cdot \mid l_{\text{img}}(t), l_{z_\mathbf{c}}, o^t(\cdot \mid z_\mathbf{c}), \mathcal{B}^t)$
6:     **end for**
7:     $j \sim p$, where $p = \prod_{c=1}^k p_c$
8:     Query grid cell with index $j$ and observe $x^{(j)}$ and true label $y^{(j)} = \{y^{(j)}(\cdot \mid z_1), \ldots, y^{(j)}(\cdot \mid z_k)\}$.
9:     Obtain $R^t = \sum_{c=1}^k y^{(j)}(\cdot \mid z_c)$; Update $o_{(j)}^t(\cdot \mid z_c)$ with $o_{(j)}^{t+1}(\cdot \mid z_c) = 2y^{(j)}(\cdot \mid z_c) - 1$ (for each $c \in \{1, \ldots, k\}$), and update $\mathcal{B}^t$ with $\mathcal{B}^{t+1} = \mathcal{B}^t - c(k, j)$ (assuming we query $k$'th grid at $(t-1)$).
10:     $R^{\text{task}} = R^{\text{task}} + R^t$; Incorporate latest observation $x^{(j)}$ into $x_{h_t}$, i.e., $x_{h_{t+1}} = \{x_{h_t}, x^{(j)}\}$.
11:     $t \leftarrow t + 1$
12: **end while**
13: **Return** $R^{\text{task}}$

---

Here, $\mathcal{Z}$ denotes the set of target categories specified in natural language (e.g., $\mathcal{Z} = \{\text{car}, \text{truck}, \text{boat}\}$), while $z_\mathbf{c}$ represents an individual category within this set. *Our proposed inference approach enables DiffVAS to flexibly handle tasks with varying numbers of target categories, overcoming a key limitation of previous VAS frameworks.* We detail our inference process in Algorithm 1.

# 4 EXPERIMENTS AND RESULTS

**Evaluation metrics.** Since VAS aims to maximize the identification of patches with target objects, we evaluate performance using the *average number of targets (ANT)* identified through exploration in partially observable environments. In this work, we focus primarily on uniform query costs, i.e., $c(i, j) = 1$ for all $i, j$, so $\mathcal{B}$ represents simply the total number of queries. Hence, ANT is defined as:

$$\text{ANT} = \frac{1}{L} \sum_{i=1}^L \sum_{t=1}^\mathcal{B} y_i^{(q_t)}(\cdot \mid \mathcal{Z}) \text{ where } L = \text{number of test search tasks instances} \quad (10)$$

We evaluate DiffVAS and baselines across varying search budgets $\mathcal{B} \in \{5, 7, 10\}$ on a $5 \times 5$ grid structure. In Appendix A2, we conduct additional experiments across various grid configurations, each employing different values of $\mathcal{B}$ with varying target category sets $\mathcal{Z}$.

**Baselines.** We compare our proposed DiffVAS policy to the following baselines:

- Random Search (RS), in which unexplored grid cells are selected uniformly at random.
- E2EVAS (Sarkar et al., 2024b), an RL-based approach for VAS in a fully observable space.
- Meta Partially Supervised VAS (MPS-VAS) (Sarkar et al., 2023), the state-of-the-art RL-based approach for single-target VAS, is designed to learn an adaptable policy in a fully observable space.

**Datasets.** We evaluate DiffVAS and the baselines on two datasets: xView (Lam et al., 2018) and DOTA (Xia et al., 2018). Both xView and DOTA are satellite image datasets, with roughly 3000 px per dimension and representing approximately 60 object categories. We use 50%, 17%, and 33% of the large satellite images to train, validate, and test the methods, respectively. In the main paper, we compare the performance of DiffVAS with the baselines using the DOTA dataset. Similar results for the xView dataset are presented in Appendix A1.

**Single-category search tasks.** We begin by considering a setting with $\mathcal{Z}$ containing a single target category, as in most prior works. We evaluate the proposed methods with the following target classes: Large Vehicle (LV), Helicopter, Ship, Plane, Roundabout, and Harbor. The results are presented in Table 1. We observe significant improvements in the performance of the proposed DiffVAS approach compared to all baselines in each different target setting, ranging from 8.9% to 28.8% improvement relative to the most competitive MPS-VAS method.

Table 1: ANT comparisons on the DOTA dataset for the single-target category setting.

| | Test with $\mathcal{Z}$ = { Ship } | | | Test with $\mathcal{Z}$ = { LV } | | | Test with $\mathcal{Z}$ = { Plane } | | |
|---|---|---|---|---|---|---|---|---|---|
| Method | $\mathcal{B} = 5$ | $\mathcal{B} = 7$ | $\mathcal{B} = 10$ | $\mathcal{B} = 5$ | $\mathcal{B} = 7$ | $\mathcal{B} = 10$ | $\mathcal{B} = 5$ | $\mathcal{B} = 7$ | $\mathcal{B} = 10$ |
| RS | 1.68 | 2.23 | 3.24 | 2.05 | 2.76 | 4.88 | 2.11 | 2.95 | 3.92 |
| E2EVAS | 1.73 | 2.47 | 3.52 | 2.19 | 3.11 | 4.91 | 2.42 | 3.14 | 4.01 |
| MPS-VAS | 1.77 | 2.50 | 3.59 | 2.22 | 3.15 | 4.96 | 2.53 | 3.17 | 4.08 |
| **DiffVAS** | **2.12** | **3.22** | **3.91** | **2.54** | **3.57** | **5.78** | **3.12** | **4.07** | **5.24** |
| | Test with $\mathcal{Z}$ = { Harbor } | | | Test with $\mathcal{Z}$ = { Roundabout } | | | Test with $\mathcal{Z}$ = { Helicopter } | | |
| Method | $\mathcal{B} = 5$ | $\mathcal{B} = 7$ | $\mathcal{B} = 10$ | $\mathcal{B} = 5$ | $\mathcal{B} = 7$ | $\mathcal{B} = 10$ | $\mathcal{B} = 5$ | $\mathcal{B} = 7$ | $\mathcal{B} = 10$ |
| RS | 1.56 | 2.43 | 3.67 | 1.54 | 2.83 | 4.04 | 1.32 | 3.15 | 4.56 |
| E2EVAS | 1.68 | 2.57 | 3.90 | 1.77 | 2.97 | 4.18 | 1.61 | 3.29 | 4.61 |
| MPS-VAS | 1.73 | 2.63 | 3.96 | 1.86 | 3.01 | 4.25 | 1.70 | 3.44 | 4.78 |
| **DiffVAS** | **2.01** | **3.15** | **4.45** | **2.32** | **3.33** | **4.89** | **2.12** | **3.91** | **5.05** |

In each target setting, search performance improves as $\mathcal{B}$ increases, with DiffVAS typically gaining a greater advantage over other baselines. As more patches are revealed, the CGM-based reconstruction becomes more accurate, allowing DiffVAS to better exploit the search space and further enhance its search policy with a larger search budget $\mathcal{B}$. The importance of TCPM is demonstrated by the superior performance of DiffVAS across all diverse target categories, as presented in Table 1.

Table 2: ANT comparisons on the DOTA dataset for the multiple-target category setting.

| | Test with $\mathcal{Z}$ = { Ship, Harbor } | | | Test with $\mathcal{Z}$ = { LV, Small Vehicle } | | | Test with $\mathcal{Z}$ = { Plane, Helicopter} | | |
|---|---|---|---|---|---|---|---|---|---|
| Method | $\mathcal{B} = 5$ | $\mathcal{B} = 7$ | $\mathcal{B} = 10$ | $\mathcal{B} = 5$ | $\mathcal{B} = 7$ | $\mathcal{B} = 10$ | $\mathcal{B} = 5$ | $\mathcal{B} = 7$ | $\mathcal{B} = 10$ |
| RS | 2.34 | 3.19 | 4.12 | 2.31 | 3.67 | 4.91 | 1.99 | 3.90 | 5.26 |
| E2EVAS | 2.37 | 3.22 | 4.14 | 2.33 | 3.71 | 4.93 | 2.04 | 3.95 | 5.30 |
| MPS-VAS | 2.38 | 3.26 | 4.18 | 2.38 | 3.72 | 4.97 | 2.09 | 3.98 | 5.33 |
| **DiffVAS** | **2.98** | **4.16** | **4.92** | **3.05** | **4.33** | **5.52** | **3.11** | **4.34** | **6.02** |

**Multi-category search tasks.** Next, we evaluate the proposed DiffVAS with $\mathcal{Z}$ encompassing multiple target categories and present the results in Table 2. We observe a substantial performance boost across various target category sets, ranging from 8.3% to 48.8% improvement relative to the most competitive baseline, highlighting the effectiveness of our proposed inference strategy. Note that, as shown in Tables 1 and 2, ANT values vary across different $\mathcal{Z}$ because each target category appears with different frequencies in the search space. Next, we analyze each module within DiffVAS.

**Importance of CGM.** To investigate the significance of CGM in the DiffVAS framework, we assess a DiffVAS variant, denoted Mask-DiffVAS, where we exclude the latent representation of the search space reconstructed using CGM (i.e., $l_{\text{re}}(t)$, cf. Fig. 3) from the input state of TCPM and compare its performance against the full DiffVAS. We see from Table 3 that DiffVAS significantly outperforms Mask-DiffVAS, with performance increases ranging from 8.1% to 37.7%. This highlights the crucial role of utilizing the latent representation of the synthesized search space $l_{\text{re}}(t)$ for planning and underscores the importance of CGM within DiffVAS.

Table 3: Significance of the conditional generative module (CGM) within DiffVAS.

| | Test with $\mathcal{Z}$ = { Ship } | | | Test with $\mathcal{Z}$ = { LV } | | | Test with $\mathcal{Z}$ = { Plane } | | |
|---|---|---|---|---|---|---|---|---|---|
| Method | $\mathcal{B}$ = 5 | $\mathcal{B}$ = 7 | $\mathcal{B}$ = 10 | $\mathcal{B}$ = 5 | $\mathcal{B}$ = 7 | $\mathcal{B}$ = 10 | $\mathcal{B}$ = 5 | $\mathcal{B}$ = 7 | $\mathcal{B}$ = 10 |
| Mask-DiffVAS | 1.82 | 2.65 | 3.29 | 2.32 | 2.91 | 4.95 | 2.45 | 3.23 | 4.03 |
| **DiffVAS** | **2.12** | **3.22** | **3.91** | **2.54** | **3.57** | **5.78** | **3.12** | **4.07** | **5.24** |
| | Test with $\mathcal{Z}$ = { Harbor } | | | Test with $\mathcal{Z}$ = { Roundabout } | | | Test with $\mathcal{Z}$ = { Helicopter } | | |
| Method | $\mathcal{B}$ = 5 | $\mathcal{B}$ = 7 | $\mathcal{B}$ = 10 | $\mathcal{B}$ = 5 | $\mathcal{B}$ = 7 | $\mathcal{B}$ = 10 | $\mathcal{B}$ = 5 | $\mathcal{B}$ = 7 | $\mathcal{B}$ = 10 |
| Mask-DiffVAS | 1.75 | 2.56 | 3.82 | 1.91 | 2.99 | 4.10 | 1.54 | 3.33 | 4.67 |
| **DiffVAS** | **2.01** | **3.15** | **4.45** | **2.32** | **3.33** | **4.89** | **2.12** | **3.91** | **5.05** |

**Importance of TCPM.** To assess the importance of the planner module in DiffVAS, we replace the TCPM with a classifier trained to predict a target-containing grid cell based on the same input state $s_t = \left( l^z_{\text{img}}(t), o^t(\cdot \mid z), \mathcal{B}^t \right)$ as the planner. The classifier is trained using binary cross-entropy loss. We then compare the performance of this modified version, *Greedy-DiffVAS*, with the original DiffVAS. We emphasize that the only distinction between Greedy-DiffVAS and DiffVAS is the replacement of the planner module with the classifier. We evaluate their performances across different target categories, as reported in Table 4. DiffVAS consistently outperforms Greedy-DiffVAS, with performance increases ranging from 16.4% to 91.0% across the various evaluation settings. These empirical results thus demonstrate that relying solely on greedy actions is inadequate for tasks that require a balance between exploration and exploitation, which highlights the critical role of the planning module in learning an efficient search policy in partially observable environments.

Table 4: Significance of the target-conditioned planning module (TCPM) within DiffVAS.

| | Test with $\mathcal{Z}$ = { Ship } | | | Test with $\mathcal{Z}$ = { LV } | | | Test with $\mathcal{Z}$ = { Plane } | | |
|---|---|---|---|---|---|---|---|---|---|
| Method | $\mathcal{B}$ = 5 | $\mathcal{B}$ = 7 | $\mathcal{B}$ = 10 | $\mathcal{B}$ = 5 | $\mathcal{B}$ = 7 | $\mathcal{B}$ = 10 | $\mathcal{B}$ = 5 | $\mathcal{B}$ = 7 | $\mathcal{B}$ = 10 |
| Greedy-DiffVAS | 1.29 | 2.01 | 2.96 | 1.81 | 2.45 | 4.46 | 2.00 | 2.57 | 3.77 |
| **DiffVAS** | **2.12** | **3.22** | **3.91** | **2.54** | **3.57** | **5.78** | **3.12** | **4.07** | **5.24** |
| | Test with $\mathcal{Z}$ = { Harbor } | | | Test with $\mathcal{Z}$ = { Roundabout } | | | Test with $\mathcal{Z}$ = { Helicopter } | | |
| Method | $\mathcal{B}$ = 5 | $\mathcal{B}$ = 7 | $\mathcal{B}$ = 10 | $\mathcal{B}$ = 5 | $\mathcal{B}$ = 7 | $\mathcal{B}$ = 10 | $\mathcal{B}$ = 5 | $\mathcal{B}$ = 7 | $\mathcal{B}$ = 10 |
| Greedy-DiffVAS | 1.23 | 2.19 | 3.32 | 1.22 | 2.57 | 3.92 | 1.11 | 3.02 | 4.34 |
| **DiffVAS** | **2.01** | **3.15** | **4.45** | **2.32** | **3.33** | **4.89** | **2.12** | **3.91** | **5.05** |

**Impact of $\mathcal{R}^{\text{GR}}$ and $\mathcal{R}^{\text{LU}}$ on search performance.** We conduct an ablation study to analyze the significance of various reward components in the proposed reward function (8). We train DiffVAS with different reward components and compare the performances across various target settings, with results reported in Table 5. The results suggest that relying solely on $\mathcal{R}^{\text{AS}}$ is insufficient, emphasizing the importance of actions that enhance information gathering about the search space. However, as would be expected, merely gathering information is not enough, as performance drops when training the policy using only $\mathcal{R}^{\text{GR}} + \mathcal{R}^{\text{LU}}$. Thus, incorporating both $\mathcal{R}^{\text{AS}}$ and $\mathcal{R}^{\text{GR}} + \mathcal{R}^{\text{LU}}$ is essential for learning an effective search policy in partially observed environments. Additionally, we observe a slight performance drop when we exclude $\mathcal{R}^{\text{LU}}$ from (8) during training.

Table 5: Ablation study of the different components of the proposed reward function.

| | Test with $\mathcal{Z}$ = { Ship } | | | Test with $\mathcal{Z}$ = { LV } | | | Test with $\mathcal{Z}$ = { Plane } | | |
|---|---|---|---|---|---|---|---|---|---|
| Reward | $\mathcal{B}$ = 5 | $\mathcal{B}$ = 7 | $\mathcal{B}$ = 10 | $\mathcal{B}$ = 5 | $\mathcal{B}$ = 7 | $\mathcal{B}$ = 10 | $\mathcal{B}$ = 5 | $\mathcal{B}$ = 7 | $\mathcal{B}$ = 10 |
| $\mathcal{R}^{\text{AS}}$ | 1.65 | 2.71 | 3.77 | 1.89 | 2.85 | 3.90 | 2.05 | 3.50 | 4.68 |
| $\mathcal{R}^{\text{GR}} + \mathcal{R}^{\text{LU}}$ | 1.63 | 2.67 | 3.66 | 1.73 | 2.78 | 3.79 | 1.80 | 3.43 | 4.69 |
| $\mathcal{R}^{\text{AS}} + \mathcal{R}^{\text{GR}}$ | 1.76 | 2.88 | 3.82 | 1.90 | 2.98 | 4.32 | 1.89 | 3.54 | 4.78 |
| **Full reward** | **2.01** | **3.15** | **4.45** | **2.32** | **3.33** | **4.89** | **2.12** | **3.91** | **5.05** |

**Effectiveness of strategy for handling multiple target categories.** We evaluate the proposed inference approach (detailed in Sec. 3.2) by comparing the performance of DiffVAS with two variants that use the same training strategy but differ in their inference methods, specifically the way $l_z$ is computed: (1) *Avg-DiffVAS* computes $l_z$ by inputting the entire target category set $\mathcal{Z}$ into the CLIP text encoder, requiring only a single forward pass through the planning module at each time step and (2) *Emb-DiffVAS* computes target-specific embeddings by processing each target category in the set $\mathcal{Z}$ individually through the CLIP text encoder, then averages them to obtain $l_z$. We compare their performance across different $\mathcal{Z}$ in Table 6. We observe that these natural alternative strategies perform worse than our proposed strategy.

Table 6: Effectiveness of the proposed inference strategy.

| | Test with $\mathcal{Z}$ = { Ship, Harbor } | | | Test with $\mathcal{Z}$ = { LV, Small Vehicle } | | | Test with $\mathcal{Z}$ = { Plane, Helicopter } | | |
| --- | --- | --- | --- | --- | --- | --- | --- | --- | --- |
| Method | $\mathcal{B}$ = 5 | $\mathcal{B}$ = 7 | $\mathcal{B}$ = 10 | $\mathcal{B}$ = 5 | $\mathcal{B}$ = 7 | $\mathcal{B}$ = 10 | $\mathcal{B}$ = 5 | $\mathcal{B}$ = 7 | $\mathcal{B}$ = 10 |
| Avg-DiffVAS | 2.45 | 3.32 | 4.45 | 2.51 | 3.82 | 5.10 | 2.21 | 4.09 | 5.55 |
| Emb-DiffVAS | 2.67 | 3.55 | 4.67 | 2.81 | 4.02 | 5.31 | 2.45 | 4.23 | 5.89 |
| **DiffVAS** | **2.98** | **4.16** | **4.92** | **3.05** | **4.33** | **5.52** | **3.11** | **4.34** | **6.02** |

**Visualizing reconstructions from the CGM.** Fig. 4 illustrates an example of CGM's reconstruction of the search space from partially observed glimpses; see more in Appendix A3.

**Zero-shot generalization.** To assess the zero-shot generalizability of DiffVAS, we evaluate a policy trained solely on DOTA, while ensuring that the target category set $\mathcal{Z}$ from DOTA differs from that in xView (this has to be done, since the categories partially overlap between these datasets). We present the result in Table 7. The results show performance improvements ranging between 36.3% to 281.5% compared to the baseline approaches and highlight the effectiveness of DiffVAS in zero-shot generalization. The superior zero-shot generalizability of DiffVAS stems from the CGM module, which preserves the strength of the trained diffusion model. This ensures that the representation extracted from CGM (i.e., $l_{re}(t)$, $l_h(t)$), a key component of the planning module's state input ($s_t$), remains robust. See Appendix A4 for additional results.

Table 7: DiffVAS has superior zero-shot generalization performance relative to the other methods.

| | Test with $\mathcal{Z}$ = { Small Car } | | | Test with $\mathcal{Z}$ = { Sail Boat } | | | Test with $\mathcal{Z}$ = { Helipad } | | |
| --- | --- | --- | --- | --- | --- | --- | --- | --- | --- |
| Method | $\mathcal{B}$ = 5 | $\mathcal{B}$ = 7 | $\mathcal{B}$ = 10 | $\mathcal{B}$ = 5 | $\mathcal{B}$ = 7 | $\mathcal{B}$ = 10 | $\mathcal{B}$ = 5 | $\mathcal{B}$ = 7 | $\mathcal{B}$ = 10 |
| E2EVAS | 1.51 | 2.03 | 3.04 | 0.25 | 0.35 | 0.47 | 0.15 | 0.21 | 0.29 |
| MPS-VAS | 1.54 | 2.09 | 3.12 | 0.27 | 0.36 | 0.49 | 0.16 | 0.31 | 0.38 |
| **DiffVAS** | **2.10** | **2.95** | **4.34** | **1.03** | **1.19** | **1.30** | **0.45** | **0.89** | **1.02** |

## 5 RELATED WORK

**Visual active search (VAS).** The VAS framework was first introduced by Sarkar et al. (2024b), who framed it as a budget-constrained MDP and tackled it using deep RL. Sarkar et al. (2023; 2024a) introduced a meta-learning approach that enables the policy to utilize supervised information gathered during the search. Key limitations of previous VAS approaches are the reliance on full observation of the search area and the focus on learning policies tailored to specific target objects, making them incapable of handling multiple target categories simultaneously. Similar to VAS, is the task of active geo-localization (Pirinen et al., 2022; Sarkar et al., 2024c), in which an agent with aerial view observations of a scene seeks to actively localize a goal. However, that task considers only the single-target location and assumes access to an observation of the target location.

**Autonomous UAV exploration.** Methodologically, our work also falls within the broad scope of literature within autonomous control and navigation of UAVs (Dang et al., 2018; Popović et al., 2020; Stache et al., 2022; Meera et al., 2019; Zhao et al., 2021; Bartolomei et al., 2020; Sadat et al., 2015). Many of these prior works (Wu et al., 2019; Yang et al., 2020; Wang et al., 2020; Thavamani et al., 2021; Meng et al., 2022a;b) assume access to a global lower-resolution observation of the whole area of interest, while DiffVAS *reconstructs* the region of interest from partial observations.

**Active scene/object reconstruction.** There is extensive prior work on active reconstruction of scenes and/or objects (Jayaraman & Grauman, 2016; 2018; Xiong & Grauman, 2018; Pirinen et al., 2019). However, these methods typically focus solely on optimizing for reconstruction, while our ultimate goal is identifying target-rich regions. Success for our task hinges on balancing *exploration* (obtaining useful information about the scene) and *exploitation* (finding objects of interest).

## 6 CONCLUSIONS

We have presented DiffVAS, a novel multi-target visual active search approach that generalizes across domains. At its core is a diffusion-based conditional generative module (CGM) that dynamically reconstructs the search area, enabling the target-conditioned planning module to plan movements effectively in a partially observable environment. Furthermore, our inference method enables DiffVAS to handle tasks that involve searching multiple target categories simultaneously, with varying category counts. Trained with a novel reward balancing exploration and exploitation, DiffVAS outperforms strong baselines and prior methods, while demonstrating excellent zero-shot generalization.

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
