# A  APPENDIX

## A.1  EVALUATION WITH XVIEW DATASET

**Evaluation in single-target category search tasks**    In this section we compare DiffVAS to baseline approaches using the xView dataset, starting with a single-target category search setting. The evaluation includes target classes such as Small Car (SC), Helicopter, Sail Boat (SB), Container Ship, Building, and Helipad. The results in Table 8 reveal a similar trend to those observed with the DOTA dataset (main paper), with significant performance improvements of the proposed DiffVAS approach over all baselines, ranging from 11.2% to 109.7% compared to the strongest baseline, MPS-VAS. The empirical results further confirm the effectiveness of our proposed DiffVAS framework in learning an efficient visual active search policy in partially observed environments.

**Evaluation in multi-target category search tasks**    In this section we evaluate DiffVAS using the xView dataset with $\mathcal{Z}$ containing multiple target categories, and the results are presented in Table 9. Consistent with our findings on the DOTA dataset (main paper), we observe a notable improvement in performance across different target category sets, ranging from 8.8% to 17.3% compared to the strongest baseline, MPS-VAS. This further underscores the effectiveness of our proposed DiffVAS inference strategy in handling diverse and complex search tasks involving multiple target categories.

Table 8: ANT comparisons on the xView dataset for the single-target category setting.

| | Test with $\mathcal{Z}$ = { Small Car } | | | Test with $\mathcal{Z}$ = { Helicopter } | | | Test with $\mathcal{Z}$ = { Bus } | | |
|---|---|---|---|---|---|---|---|---|---|
| Method | $\mathcal{B}$ = 5 | $\mathcal{B}$ = 7 | $\mathcal{B}$ = 10 | $\mathcal{B}$ = 5 | $\mathcal{B}$ = 7 | $\mathcal{B}$ = 10 | $\mathcal{B}$ = 5 | $\mathcal{B}$ = 7 | $\mathcal{B}$ = 10 |
| RS | 1.92 | 2.51 | 3.51 | 0.17 | 0.24 | 0.41 | 0.31 | 0.35 | 0.48 |
| E2EVAS | 2.37 | 3.07 | 3.88 | 0.19 | 0.28 | 0.40 | 0.29 | 0.41 | 0.47 |
| MPS-VAS | 2.45 | 3.12 | 3.93 | 0.23 | 0.31 | 0.43 | 0.30 | 0.43 | 0.51 |
| **DiffVAS** | **2.91** | **3.89** | **4.53** | **0.45** | **0.65** | **0.81** | **0.42** | **0.56** | **0.66** |
| | Test with $\mathcal{Z}$ = { Building } | | | Test with $\mathcal{Z}$ = { Container Ship } | | | Test with $\mathcal{Z}$ = { Truck } | | |
| Method | $\mathcal{B}$ = 5 | $\mathcal{B}$ = 7 | $\mathcal{B}$ = 10 | $\mathcal{B}$ = 5 | $\mathcal{B}$ = 7 | $\mathcal{B}$ = 10 | $\mathcal{B}$ = 5 | $\mathcal{B}$ = 7 | $\mathcal{B}$ = 10 |
| RS | 2.34 | 3.32 | 3.91 | 0.20 | 0.31 | 0.42 | 0.18 | 0.31 | 0.40 |
| E2EVAS | 2.61 | 3.45 | 4.18 | 0.21 | 0.35 | 0.47 | 0.18 | 0.33 | 0.42 |
| MPS-VAS | 2.68 | 3.51 | 4.22 | 0.23 | 0.36 | 0.48 | 0.21 | 0.37 | 0.45 |
| **DiffVAS** | **2.93** | **3.92** | **4.52** | **0.31** | **0.45** | **0.60** | **0.29** | **0.50** | **0.62** |

Table 9: ANT comparisons on the xView dataset for the multiple-target category setting.

| | Test with $\mathcal{Z}$ = { Small Car, Bus } | | | Test with $\mathcal{Z}$ = { Small Car, Truck } | | | Test with $\mathcal{Z}$ = { Small Car, Building } | | |
|---|---|---|---|---|---|---|---|---|---|
| Method | $\mathcal{B}$ = 5 | $\mathcal{B}$ = 7 | $\mathcal{B}$ = 10 | $\mathcal{B}$ = 5 | $\mathcal{B}$ = 7 | $\mathcal{B}$ = 10 | $\mathcal{B}$ = 5 | $\mathcal{B}$ = 7 | $\mathcal{B}$ = 10 |
| RS | 2.12 | 2.67 | 3.31 | 2.01 | 2.89 | 3.77 | 2.53 | 3.91 | 5.44 |
| E2EVAS | 2.46 | 3.25 | 4.05 | 2.43 | 3.17 | 3.96 | 2.97 | 4.66 | 5.31 |
| MPS-VAS | 2.53 | 3.32 | 4.11 | 2.48 | 3.25 | 4.03 | 3.08 | 4.95 | 5.45 |
| **DiffVAS** | **3.02** | **3.81** | **4.54** | **2.91** | **3.99** | **4.46** | **3.67** | **5.32** | **5.93** |

## A.2  EVALUATION WITH DIFFERENT GRID SIZES

We evaluate DiffVAS performance on a $10 \times 10$ grid using the DOTA dataset, with results shown in Table 10. Table 10 shows results for the single-target category search task. We compare the search performance of the proposed approach against the baselines across different search budgets, $\mathcal{B} \in \{25, 30, 35\}$, on a $10 \times 10$ grid. Similar to the $5 \times 5$ setting, we observe a performance improvement over the baselines, ranging from 0.1% to 8.8% across different evaluation scenarios. As anticipated, the overall performance is lower in the larger grid setting, highlighting the increased difficulty and motivating further research into VAS in partially observable environments.

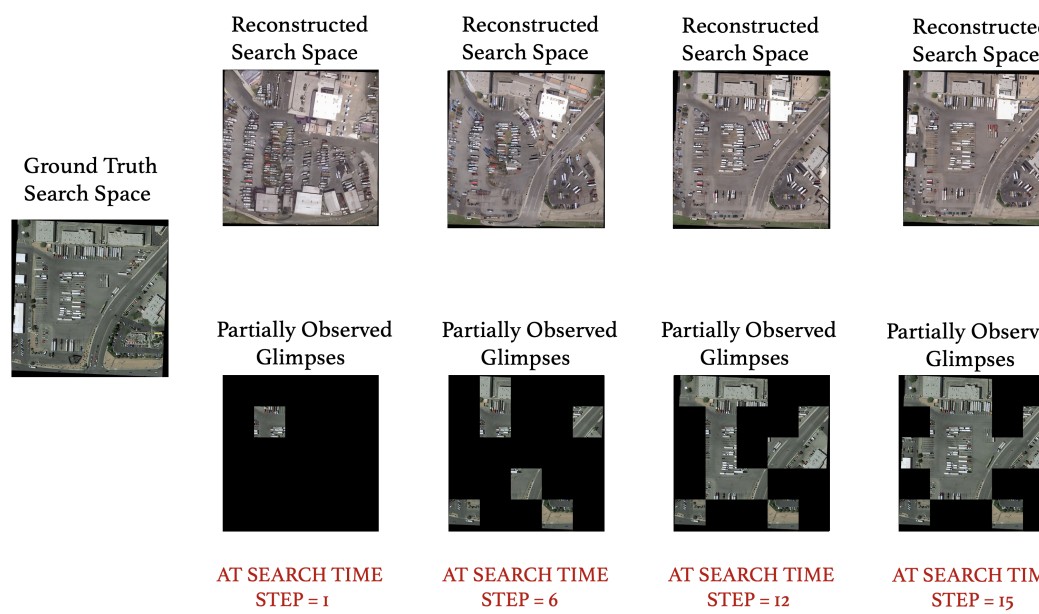

Figure 5: Visualizations of CGM's reconstruction of the search space from partially observed glimpses at various stages of the search. The reconstruction quality improves as more patches are revealed.

Table 10: ANT comparisons on the DOTA dataset for the single-target category in $10 \times 10$ settings.

| | Test with $\mathcal{Z} = \{$ Large Vehicle $\}$ | | | Test with $\mathcal{Z} = \{$ Helicopter $\}$ | | | Test with $\mathcal{Z} = \{$ Plane $\}$ | | |
|---|---|---|---|---|---|---|---|---|---|
| Method | $\mathcal{B} = 25$ | $\mathcal{B} = 30$ | $\mathcal{B} = 35$ | $\mathcal{B} = 25$ | $\mathcal{B} = 30$ | $\mathcal{B} = 35$ | $\mathcal{B} = 25$ | $\mathcal{B} = 30$ | $\mathcal{B} = 35$ |
| RS | 5.92 | 7.12 | 8.21 | 1.19 | 1.32 | 1.41 | 5.32 | 7.02 | 8.19 |
| E2EVAS | 6.34 | 7.74 | 8.91 | 1.27 | 1.44 | 1.56 | 5.87 | 7.74 | 8.92 |
| MPS-VAS | 6.37 | 7.80 | 8.96 | 1.32 | 1.49 | 1.60 | 5.93 | 7.83 | 9.01 |
| **DiffVAS** | **6.39** | **7.94** | **9.02** | **1.39** | **1.55** | **1.74** | **6.12** | **7.99** | **9.21** |
| | Test with $\mathcal{Z} = \{$ Roundabout $\}$ | | | Test with $\mathcal{Z} = \{$ Ship $\}$ | | | Test with $\mathcal{Z} = \{$ Harbor $\}$ | | |
| Method | $\mathcal{B} = 25$ | $\mathcal{B} = 30$ | $\mathcal{B} = 35$ | $\mathcal{B} = 25$ | $\mathcal{B} = 30$ | $\mathcal{B} = 35$ | $\mathcal{B} = 25$ | $\mathcal{B} = 30$ | $\mathcal{B} = 35$ |
| RS | 5.11 | 7.02 | 8.13 | 1.10 | 1.15 | 1.47 | 5.47 | 7.02 | 8.57 |
| E2EVAS | 5.75 | 7.63 | 8.69 | 4.78 | 5.96 | 7.88 | 6.15 | 7.98 | 9.24 |
| MPS-VAS | 5.82 | 7.71 | 8.78 | 4.83 | 6.04 | 7.97 | 6.19 | 8.03 | 9.35 |
| **DiffVAS** | **5.94** | **7.92** | **8.89** | **5.05** | **6.23** | **8.09** | **6.31** | **8.32** | **9.56** |

## A.3 VISUALIZATIONS OF CGM SEARCH SPACE RECONSTRUCTIONS FROM PARTIALLY OBSERVED GLIMPSES

In this section we provide additional illustrative visualizations of CGM's reconstruction of search spaces from partially observed glimpses at various stages of the search, corresponding to different history lengths ($h_t$). These visualizations are obtained using the CGM trained with the DOTA dataset. We depict the visualizations in Fig. 5, 6 and 7. These visualizations offer a qualitative perspective on CGM's search space reconstruction quality derived from partially observed glimpses.

## A.4 IMPLEMENTATION DETAILS

In this section, we detail the training process for DiffVAS. The proposed DiffVAS framework comprises two modules: the conditional generative module (CGM), and the task-conditioned planning module (TCPM). Since each module is trained independently, we discuss the training details for each module separately, beginning with the CGM.

**Details of CGM** We use Stable Diffusion v2.1 (Rombach et al., 2022) as CGM's primary latent diffusion model. We integrate partially observed glimpses into the diffusion model by attaching a trainable adapter module following (Zhang et al., 2023). The diffusion model[2] is kept frozen while

---

[2]https://huggingface.co/stabilityai/stable-diffusion-2-1

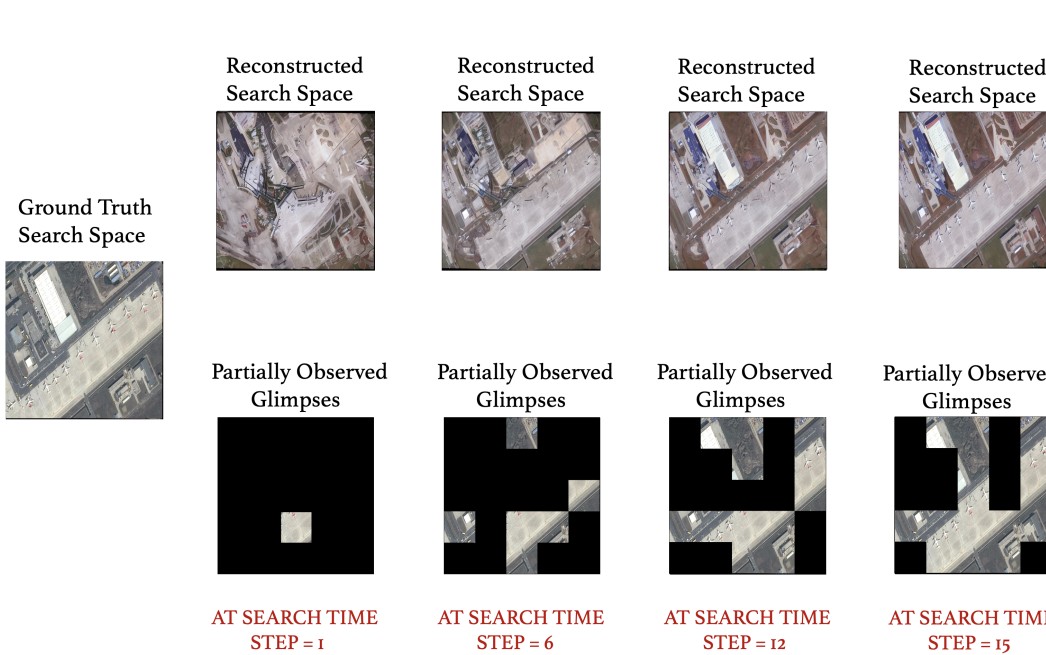

Figure 6: Visualizations of CGM's reconstruction of the search space from partially observed glimpses at various stages of the search. The reconstruction quality improves as more patches are revealed.

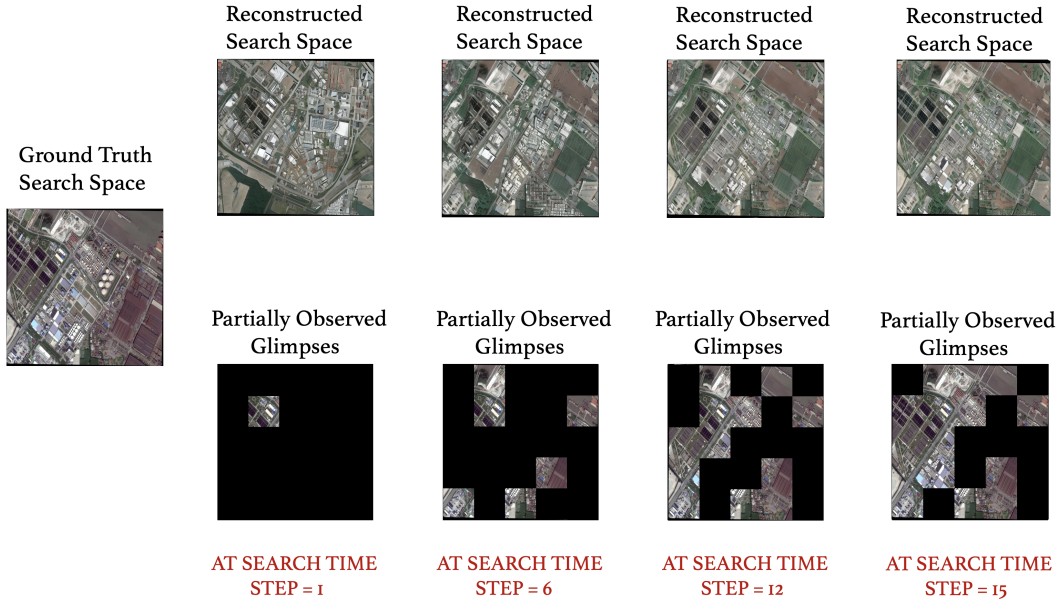

Figure 7: Visualizations of CGM's reconstruction of the search space from partially observed glimpses at various stages of the search. The reconstruction quality improves as more patches are revealed.

the adapter module is fully optimized end-to-end. We use an empty string as the input prompt to the latent diffusion model during training and inference. We randomly mask the ground truth image during training and pass it to the adapter module. We use the Adam optimizer (Kingma, 2014) and a learning rate of 1e-5 to optimize the adapter module.

**Details of TCPM** The planner module consists of four main components: (1) a CLIP text encoder (Radford et al., 2021); (2) a cross-attention module; (3) a positional encoding module; and (4) an actor-critic network with a shared backbone. The CLIP text encoder provides information about the target category to the planner module, by computing the text embedding of the target category. The cross-attention module consists of a single cross-attention block [3] that fuses the information from the CGM and the CLIP text encoder. The positional encoding module provides TCPM with two key pieces of information: (i) the relative positions of each patch in the latent representation space; and (ii) the positions of the revealed patches, and the position of the patches that CGM has reconstructed. Finally, the actor and critic network consists of a lightweight shared backbone comprising 3 convolutional filters and pooling operations. The actor head and critic head are simple MLPs, each comprising three hidden layers with Tanh non-linear activation layers in between. We also incorporate a softmax activation at the final layer of the actor network to output a probability distribution over the grid cells. Except for the CLIP text encoder [4], all the components of the TCPM are trainable. We use a learning rate of 1e-4, batch size of 1, number of training steps as 100,000, and the Adam optimizer. The script for training and inference of DiffVAS can be accessed through the anonymous link provided here.

**Compute resources** We use a single NVidia H100 GPU server with a memory of 80 GB for training and a single NVidia V100 GPU server with a memory of 32 GB for running the inference. It requires approximately 50 GPU hours to train TCPM, while the adapter module is optimized for approximately 100 GPU hours. The inference time is 22 seconds for a single search task on a single NVidia V100 GPU, with a maximum exploration budget $\mathcal{B}$ of 10. Precisely, our end-to-end DiffVAS framework infers the next region to query in approximately 2.20 seconds on a standard NVIDIA V100 GPU. Specifically, a diffusion-based CGM module approximately takes 1.07 seconds to reconstruct a single full image on NVIDIA V100 GPU with 32GB of GPU memory. Given that verifying a search query by a park ranger typically takes a few minutes to hours (depending on the search space), the response time of our system is well within the operational requirements. This efficiency ensures that our framework can provide timely and actionable support, making it highly suitable for real-world deployment where swift decision-making is crucial.

## A.5 VISUALIZATION OF EXPLORATION BEHAVIOR OF DIFFVAS

In this section, we showcase visualizations of exploration behaviors of DiffVAS across diverse search tasks, covering both single- and multi-target category searches. These visualizations are obtained using the DiffVAS policy trained with the DOTA dataset. We depict the visualizations in Fig. 8, 9 and 10. These visualizations provide a comprehensive view of the exploration behaviors of DiffVAS, enabling a nuanced comparison of how the policy adapts to different target specifications.

## A.6 EFFECTIVENESS OF CROSS-ATTENTION LAYER

Here, we analyze the efficacy of the cross-attention layer in learning a target-aware representation suitable for planning. To evaluate the role of the cross-attention layer in the target-conditioned planner module, we remove it and instead concatenate $l_{\text{img}}(t)$ and $l_z$ channel-wise to derive the target-aware representation $l_{\text{img}}^z(t)$, while keeping the rest of the DiffVAS framework unchanged. We refer to the resulting framework as *Concat-DiffVAS*. We compare the performance of DiffVAS and Concat-DiffVAS across different target category sets $\mathcal{Z}$ using the DOTA dataset, as shown in Table 11. We see that in most cases, Concat-DiffVAS obtains a lower ANT score, which indicates that the cross-attention layer is effective in learning target-aware representations for planning.

---

[3] https://style-aligned-gen.github.io/

[4] https://huggingface.co/openai/clip-vit-base-patch32

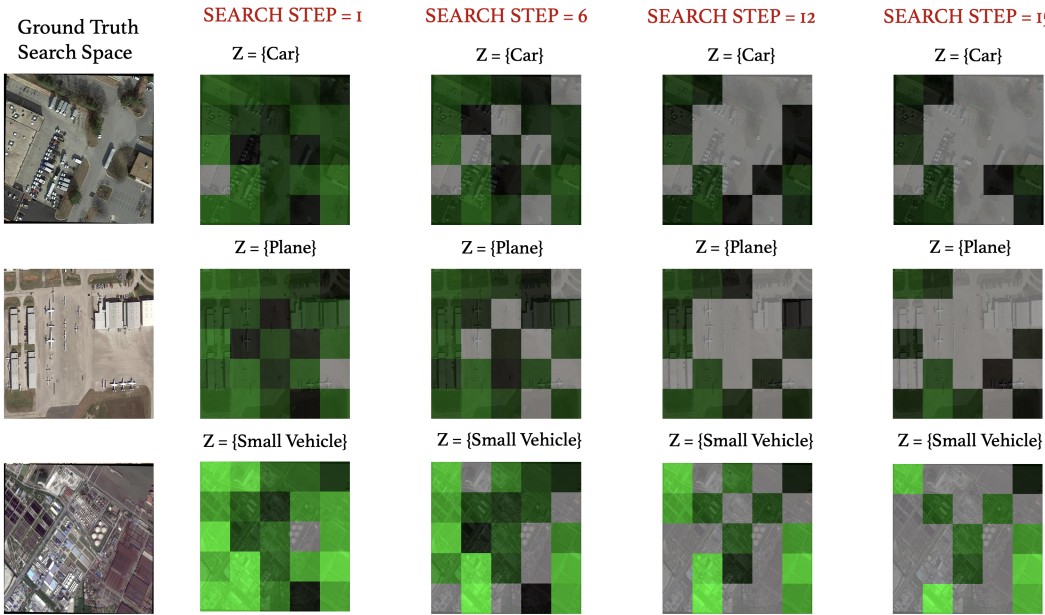

Figure 8: Query sequences for different target category sets $\mathcal{Z}$, as well as corresponding heat maps (darker indicates higher probability). Note that as the search proceeds, the agent becomes relatively more confident (lower entropy) in terms of where it wants to query next.

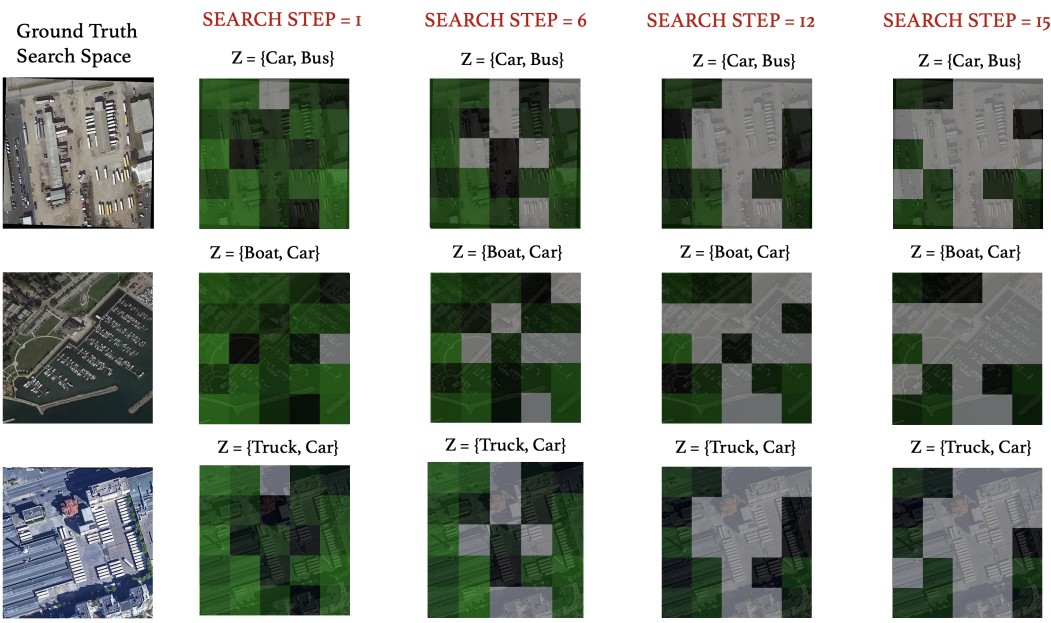

Figure 9: Query sequences for different target category sets $\mathcal{Z}$, as well as corresponding heat maps (darker indicates higher probability). Note that as the search proceeds, the agent becomes relatively more confident (lower entropy) in terms of where it wants to query next.

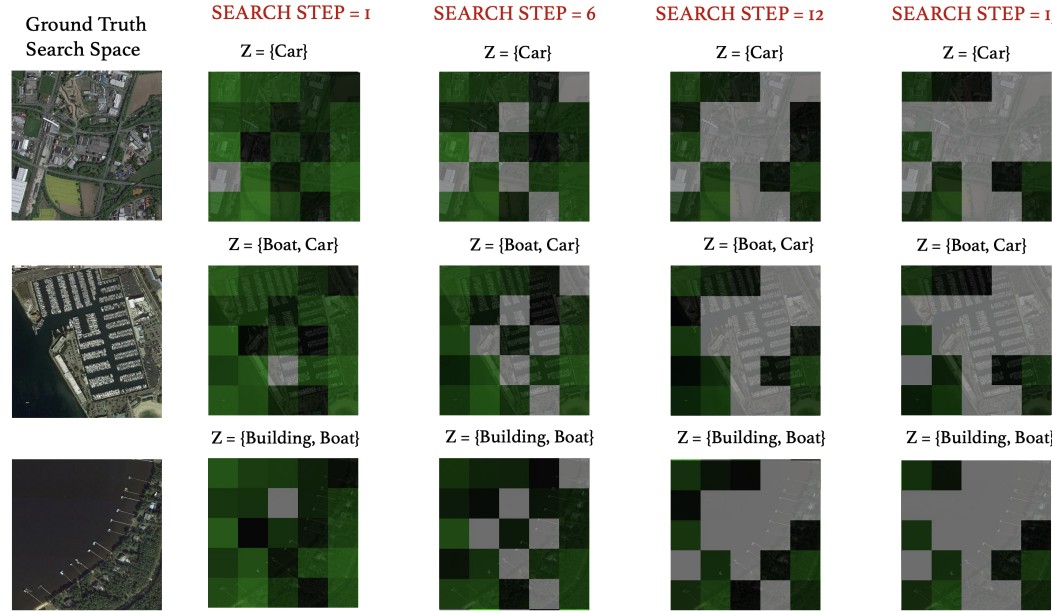

Figure 10: Query sequences for different target category sets $\mathcal{Z}$, as well as corresponding heat maps (darker indicates higher probability). Note that as the search proceeds, the agent becomes relatively more confident (lower entropy) in terms of where it wants to query next.

Table 11: Effectiveness of cross-attention layer in learning a target-aware representation.

| | Test with $\mathcal{Z}$ = { Large Vehicle } | | | Test with $\mathcal{Z}$ = { Helicopter } | | | Test with $\mathcal{Z}$ = { Plane } | | |
|---|---|---|---|---|---|---|---|---|---|
| Method | $\mathcal{B} = 5$ | $\mathcal{B} = 7$ | $\mathcal{B} = 10$ | $\mathcal{B} = 5$ | $\mathcal{B} = 7$ | $\mathcal{B} = 10$ | $\mathcal{B} = 5$ | $\mathcal{B} = 7$ | $\mathcal{B} = 10$ |
| Concat-DiffVAS | 2.02 | 3.13 | **4.21** | 2.36 | 3.41 | **5.94** | 3.04 | 3.92 | 5.12 |
| **DiffVAS** | **2.12** | **3.22** | 3.91 | **2.54** | **3.57** | 5.78 | **3.12** | **4.07** | **5.24** |
| | Test with $\mathcal{Z}$ = { Roundabout } | | | Test with $\mathcal{Z}$ = { Ship } | | | Test with $\mathcal{Z}$ = { Harbor } | | |
| Method | $\mathcal{B} = 5$ | $\mathcal{B} = 7$ | $\mathcal{B} = 10$ | $\mathcal{B} = 5$ | $\mathcal{B} = 7$ | $\mathcal{B} = 10$ | $\mathcal{B} = 5$ | $\mathcal{B} = 7$ | $\mathcal{B} = 10$ |
| Concat-DiffVAS | 1.91 | 3.08 | 4.42 | 2.30 | 3.27 | **4.95** | 2.04 | 3.80 | **5.11** |
| **DiffVAS** | **2.01** | **3.15** | **4.45** | **2.32** | **3.33** | 4.89 | **2.12** | **3.91** | 5.05 |

## A.7 COMPARISON WITH FULLY OBSERVABLE SEARCH SETTING

To evaluate the efficacy of the proposed DiffVAS framework, we compare its performance to a similar approach that assumes full observability of the search space, referred to as *FullVAS*. FullVAS is identical to DiffVAS, except it provides the full search space image $x$ to the $e^{\text{CGM}}$ feature extractor to derive the latent representation of search space (denoted as $l_{\text{img}}$), i.e., $l_{\text{img}}(t) = e^{\text{CGM}}(x)$. Interestingly, we see from Table 12 that DiffVAS achieves results comparable to those of FullVAS, despite the fact that DiffVAS never observes the full search area as FullVAS does. This further showcases the strength of our proposed approach, and highlights the strong benefit of the diffusion-based CGM module which reconstructs the underlying search area on the fly.

Table 12: DiffVAS achieves results that are on average very close to those of FullVAS, despite the fact that DiffVAS never observes the entire search area (which FullVAS does).

| | Test with $\mathcal{Z}$ = { Large Vehicle } | | | Test with $\mathcal{Z}$ = { Helicopter } | | | Test with $\mathcal{Z}$ = { Plane } | | |
|---|---|---|---|---|---|---|---|---|---|
| Method | $\mathcal{B} = 5$ | $\mathcal{B} = 7$ | $\mathcal{B} = 10$ | $\mathcal{B} = 5$ | $\mathcal{B} = 7$ | $\mathcal{B} = 10$ | $\mathcal{B} = 5$ | $\mathcal{B} = 7$ | $\mathcal{B} = 10$ |
| FullVAS | **2.31** | **3.32** | 3.87 | **2.62** | **3.76** | **5.89** | **3.22** | **4.19** | 5.19 |
| **DiffVAS** | 2.12 | 3.22 | **3.91** | 2.54 | 3.57 | 5.78 | 3.12 | 4.07 | **5.24** |
| | Test with $\mathcal{Z}$ = { Roundabout } | | | Test with $\mathcal{Z}$ = { Ship } | | | Test with $\mathcal{Z}$ = { Harbor } | | |
| Method | $\mathcal{B} = 5$ | $\mathcal{B} = 7$ | $\mathcal{B} = 10$ | $\mathcal{B} = 5$ | $\mathcal{B} = 7$ | $\mathcal{B} = 10$ | $\mathcal{B} = 5$ | $\mathcal{B} = 7$ | $\mathcal{B} = 10$ |
| FullVAS | **2.20** | **3.31** | 4.43 | **2.35** | **3.41** | **4.95** | **2.22** | **4.03** | **5.15** |
| **DiffVAS** | 2.01 | 3.15 | **4.45** | 2.32 | 3.33 | 4.89 | 2.12 | 3.91 | 5.05 |

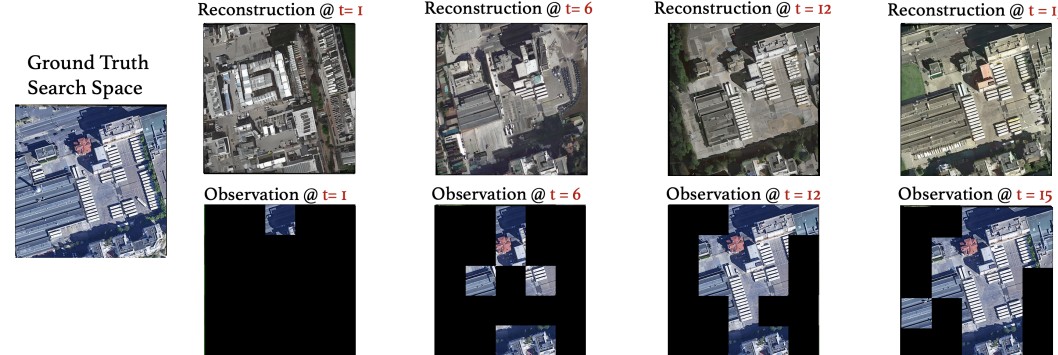

Figure 11: Visualizations of CGM's reconstruction of the search space from partially observed glimpses at various stages of the search. The reconstruction quality improves as more patches are revealed.

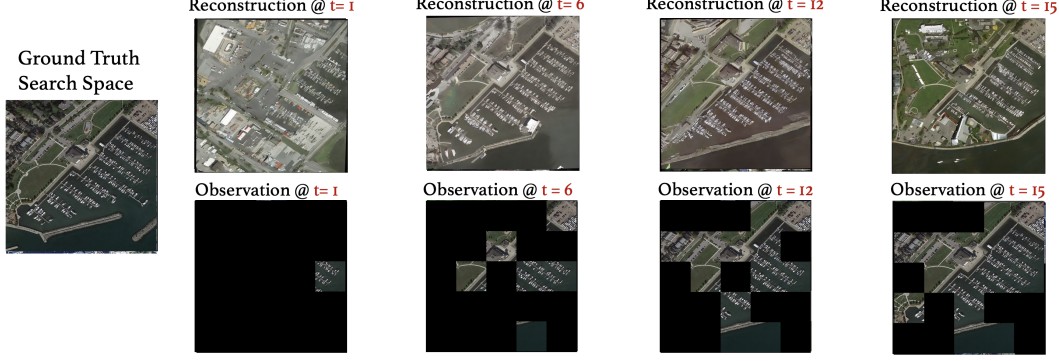

Figure 12: Visualizations of CGM's reconstruction of the search space from partially observed glimpses at various stages of the search. The reconstruction quality improves as more patches are revealed.

## A.8 MORE VISUALIZATIONS OF CGM SEARCH SPACE RECONSTRUCTIONS FROM PARTIALLY OBSERVED GLIMPSES

In this section, we present additional visualizations of search space reconstruction by CGM from partially observed glimpses at different stages of the search. See figure 11, 12, 13.

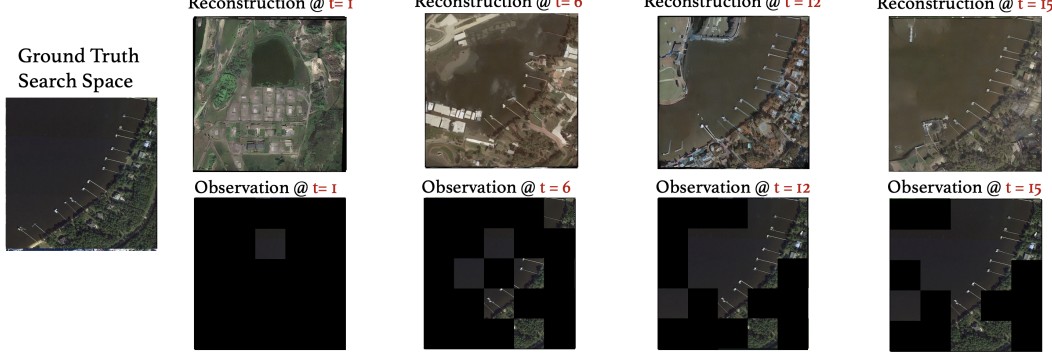

Figure 13: Visualizations of CGM's reconstruction of the search space from partially observed glimpses at various stages of the search. The reconstruction quality improves as more patches are revealed.

### A.9 REASONING FOR SELECTING THE TARGET CATEGORIES FOR EVALUATION

In both xView and DOTA, certain classes are extremely rare within the dataset. For instance, images containing at least one instance of class Cement-Mixer appear only once. Consequently, including such classes as targets is not meaningful, as evaluating performance based on a single image does not provide robust analysis/results. Therefore, we excluded these classes. A similar situation applies to multi-target categories. This consideration forms the primary motivation behind our selection of target classes. Note that for zero-shot evaluation on xView, we train models on DOTA using categories that do not appear in xView.

### A.10 PROCEDURE FOR SAMPLING AN EPISODE DURING TRAINING

During training, we randomly sample an image from the training set and select a random budget within the range {1 to N-1} (where N is the number of grids). We then extract the set of target categories if at least one instance of each category is present in the image, based on the ground truth annotations (which also include the precise locations of these target objects as a bounding box). Once the target category set is determined, we randomly select a category from this set to train our policy. The rationale for using a random budget and random target is to ensure that the policy learns to be both target and budget-agnostic. Across various experiments, we demonstrate that DiffVAS consistently outperforms the baseline across different budgets and target categories.

### A.11 EFFICACY OF LOCAL UNCERTAINTY BASED REWARD $\mathcal{R}^{\text{LU}}$

Table 13: Effectiveness of $\mathcal{R}^{\text{LU}}$.

| Reward | Test with $\mathcal{Z}$ = { Ship } | | | Test with $\mathcal{Z}$ = { LV } | | | Test with $\mathcal{Z}$ = { Plane } | | |
|---|---|---|---|---|---|---|---|---|---|
| | $\mathcal{B}$ = 5 | $\mathcal{B}$ = 7 | $\mathcal{B}$ = 10 | $\mathcal{B}$ = 5 | $\mathcal{B}$ = 7 | $\mathcal{B}$ = 10 | $\mathcal{B}$ = 5 | $\mathcal{B}$ = 7 | $\mathcal{B}$ = 10 |
| $\mathcal{R}^{\text{AS}}$ | 1.65 | 2.71 | 3.77 | 1.89 | 2.85 | 3.90 | 2.05 | 3.50 | 4.68 |
| $\mathcal{R}^{\text{AS}} + \mathcal{R}^{\text{LU}}$ | **1.71** | **2.79** | **3.79** | **1.90** | **2.92** | **4.11** | **2.09** | **3.53** | **4.74** |

In order to analyze the importance of $\mathcal{R}^{\text{LU}}$, we conduct an additional experiment where we trained DiffVAS using the $R^{AS} + R^{LU}$ reward function and compare its performance with the reward function containing only $\mathcal{R}^{\text{AS}}$. We observed a slight improvement in search performance when incorporating $R^{LU}$ into the reward function for training the TCPM module, underscoring the importance of the local uncertainty-based reward factor ($R^{LU}$).

### A.12 FURTHER DETAILS OF CROSS-ATTENTION LAYER

We depict the cross-attention layer in Figure 3. Note that, by "reference features", we refer to the latent features extracted by the CGM module, specifically the latent representation of the reconstructed image (i.e., $l_{img}(t)$). On the other hand, "target features" refer to the latent features of the target category, which are computed using the CLIP-based text encoder (i.e., $l_z$). Additionally, "AdaIN" stands for "adaptive instance normalization", as originally proposed in Huang & Belongie (2017).