# OpenReview forum: "DiffVAS: Diffusion-Guided Visual Active Search in Partially Observable Environments"
_ICLR.cc/2025/Conference — Submitted to ICLR 2025_

### Official Review · Reviewer_BQ2P · 2024-10-30

**Soundness:** 2
**Presentation:** 3
**Contribution:** 2
**Rating:** 5
**Confidence:** 4

**Summary:**

This paper introduces DiffVAS, a Visual Active Search (VAS) framework designed to handle multi-target search tasks in partially observable environments. The authors fine-tune a diffusion-based Conditional Generative Module (CGM) to reconstruct the full scene from partial observations then train a target-conditioned planning module (TCPM) using reinforcement learning with an exploration-exploition reward system to guide search policies based on target categories

**Strengths:**

1. The integration of a diffusion model for scene reconstruction within a VAS framework is novel and enables DiffVAS to work effectively with partial observations, a realistic approach for UAV-based geospatial searches.
2. DiffVAS shows good performance on both single- and multi-target tasks.

**Weaknesses:**

1. The diffusion model, while powerful, adds computational complexity. Real-time search and UAV deployment may be challenging
2. While cross-attention is introduced in DiffVAS framework, its specific contribution to performance is only briefly mentioned.  A deeper analysis might clarify its role further.
3. The evaluation relies primarily on the "Average Number of Targets" (ANT) metric, which, while informative, does not capture all aspects of performance, especially in practical scenarios. Metrics reflecting search efficiency (e.g., path length or cost-effectiveness) could provide a more holistic assessment
4. The paper simplifies the search task significantly by assuming a fixed UAV altitude and discretizing the action space into a grid structure. This abstraction does not capture the full complexity of real-world search tasks, where UAVs may need to operate at varying altitudes and adjust scale dynamically. The authors could explore multi-scale Active Visual Search (AVS) environments to better simulate realistic search scenarios.
5. By focusing solely on the exploration-exploitation balance through reinforcement learning, the framework oversimplifies the role of object detection. In practical applications, detection often involves noise, occlusions, and variable scene complexities, which are currently unaddressed. Future work could explore more challenging detection settings to evaluate DiffVAS’s resilience under realistic conditions.

**Questions:**

1. DiffVAS leverages a diffusion model to reconstruct the scene from partial observations, which could introduce significant computational overhead, especially in real-time or resource-constrained scenarios. Could the authors clarify the approximate time required for the diffusion model to reconstruct a single full observation image?
2. Can the authors provide further explanation and ablation studies or visualizations about the cross-attention layer in Figure 3?  What's the reference features and the target features? What's the AdaIN?
3. To better assess the contribution of the Conditional Generative Module (CGM), would it be feasible to input the ground truth full observation directly to the Target-Conditioned Planning Module (TCPM) to establish an upper bound on TCPM’s performance? This experiment could reveal TCPM's ideal capabilities and help quantify the effectiveness of CGM in assisting TCPM under varying levels of partial observations.

---

> ### Author Response · Authors · 2024-11-14
>
> Thanks, we appreciate the reviewer's valuable feedback! We've included all the additional details as requested.
>
> > **Q1** Why "ANT" metric?
>
> **A1** Thank you for the suggestion. Exploration is crucial for solving this task, making it possible to optimize a policy for higher ANT at the expense of increased path length. Since our primary focus is maximizing target object discovery within a given budget, we evaluate performance using ANT. In scenarios where movement cost outweighs acquisition cost, path length would be a more suitable metric. However, in our work, we assume movement cost is negligible compared to acquisition cost (for example, acquisition cost might include sending a team on the ground for verification or having a human analyst review imagery), a premise we believe is practical in most cases.
>
> > **Q2** Multi-scale AVS to simulate realistic search scenarios?
>
> **A2** To the best of our knowledge, this is the first attempt to address Visual Active Search in Partially Observable Environments. Our work is designed to serve as a foundational contribution to the emerging field of "Active Search in Partially Observable Environments." We believe this pioneering approach will pave the way for numerous future advancements in this important area. We are truly excited about the prospect of exploring this suggested direction in future work, as we agree that incorporating multi-scale AVS would bring our approach even closer to real-world deployability. Note, however, that there are also real-world scenarios where a UAV may have to hover at a fixed altitude -- for example, in a hurricane-hit area with strong winds, flying at a constant and quite low altitude may be necessary to avoid heavier winds higher up.
>
> > **Q3** Future work could explore more challenging detection settings to evaluate DiffVAS’s resilience under realistic conditions.
>
> **A3** We would like to emphasize that DiffVAS is evaluated across a wide range of target object categories and diverse scenes, showcasing its robustness. Additionally, integrating an off-the-shelf object detection module could further enhance DiffVAS to address practical challenges like occlusions, noise, and other real-world obstacles. We agree with you that this could be explored in future work, as our current effort serves as a foundational contribution to the emerging field of "Active Search in Partially Observable Environments," with the potential to drive future advancements in handling more complex real-world detection challenges.
>
> > **Q4** Computational overhead of DiffVAS? Approximate time required for the diffusion model to reconstruct a full image?
>
> **A4** We acknowledge the additional computational complexity introduced by the diffusion-based CGM module. However, our end-to-end DiffVAS framework infers the next region to query in approximately **2.20 seconds** on a standard GPU. Specifically, a diffusion-based CGM module approximately takes **1.07** seconds to reconstruct a single full image on NVIDIA V100 GPU. Given that verifying a search query on the ground by a park ranger typically takes much longer (in the order of minutes to hours), the response time of our system is well within the operational requirements. This efficiency ensures that our framework can provide timely and actionable support, making it highly suitable for real-world deployment where swift decision-making is crucial. **These additional details have been included in Appendix A9 of the updated draft.**
>
> > **Q5** (1) Further explanation about the cross-attention layer in Fig.3? (2) What\'s the reference and the target features? What's AdaIN?
>
> **A5** (1) **We had already analyzed the suggested ablation study in our original submission. We've analyzed the efficacy of the cross-attention layer and reported our findings in Appendix A6, titled "Effectiveness of Cross-Attention Layer".** Our findings indicate that the cross-attention layer is effective in learning target-aware representation.
>
> (2) **By "reference features," we refer to the latent features extracted by the CGM module, specifically the latent representation of the reconstructed image (i.e., $l_{img}(t)$). On the other hand, "target features" refer to the latent features of the target category, which are computed using the CLIP-based text encoder (i.e.,$l_{z}$)**. Additionally, "AdaIN" stands for "adaptive instance normalization," as originally proposed in [1]. We apologize for omitting these details. These additional details have been included in Appendix A12 of the updated draft.
>
> > **Q6** Assessing the contribution of CGM?
>
> **A6**  **We previously performed the suggested ablation study in our original submission, and the details are provided in Appendix A7.** The empirical results clearly demonstrate the significant advantage of the diffusion-based CGM module, which reconstructs the search area on the fly.
>
> **References:**
> 1. Arbitrary Style Transfer in Real-Time With Adaptive Instance Normalization; Xun Huang et. al.; ICCV, 2017

---

> > ### Author Response · Authors · 2024-12-01
> > **Reminder to Reviewer BQ2P**
> >
> > Dear Reviewer BQ2P,
> >
> > As the discussion period approaches its conclusion, we would like to follow up to ensure that all your concerns have been fully addressed. If you require any further clarifications, please do not hesitate to let us know, and we will be glad to provide them promptly. Thank you for all your time and effort.
> >
> > Regards,
> >
> > Authors

---

### Official Review · Reviewer_uags · 2024-10-31

**Soundness:** 3
**Presentation:** 3
**Contribution:** 3
**Rating:** 6
**Confidence:** 4

**Summary:**

The paper presents DiffVAS, a framework for visual active search in partially observable environments. It introduces a new task setup, TC-POVAS, and uses a Conditional Generative Module (CGM) and a Target-Conditioned Planning Module (TCPM) to balance exploration and exploitation. DiffVAS achieves superior performance in multi-target search tasks through extensive experiments. However, there are some issues like an incomplete ablation study on reward functions, writing inconsistencies, and formatting errors in figures and tables.

**Strengths:**

1. The introduction of a Target-Conditioned Partially Observable Environment (TC-POVAS) enables multi-target visual active search to operate in partially observable environments, making it more aligned with real-world scenarios.
2. By utilizing a Conditional Generative Module (CGM) based on a diffusion model, DiffVAS can reconstruct the entire search space from partial observations, providing more accurate guidance for subsequent search decisions.
3. The introduction of the Target-Conditioned Planning Module (TCPM) uses a reinforcement learning strategy to balance exploration and exploitation, optimizing search performance in partially observable environments.
4. Through extensive experiments on multiple public satellite imagery datasets such as xView and DOTA, DiffVAS demonstrated superior performance in multi-target search tasks, achieving significant improvements over existing methods.

**Weaknesses:**

1. The paper’s ablation study on the reward function lacks the combination of $R^{AS} + R^{LU}$, which could provide deeper insights into the impact of these components on model performance.
2. There are writing issues such as inconsistent capitalization of symbols like "Sgn" in the formulas (6) and (7).
3. Figure 3 has issues related to font, capitalization, and symbol representation. There is also a missing parenthesis in "Past Query Outcomes," which affects the clarity and professionalism of the presentation.
4. In the "Zero-shot Generalization" section, there is a misexpression where it states "solely on DOTA on xView".

**Questions:**

1. Why was the combination of $R^{AS} + R^{LU}$ omitted in the reward function ablation study?
2. Is the presence of issues in writing and formatting, such as text descriptions and figure/table formats, an indication that the paper was prepared somewhat hastily?

---

> ### Author Response · Authors · 2024-11-14
>
> Thank you for reviewing our paper! We address all your concerns below.
>
>
> > **Q1**: There are writing issues such as inconsistent capitalization of symbols like "Sgn" in the formulas (6) and (7).
>
> **A1**: Thank you for pointing this out. **We have fixed it in the updated version**, and will carefully go over the rest of the style and formats to be consistent everywhere.
>
> > **Q2**: Figure 3 has issues related to font, capitalization, and symbol representation. There is also a missing parenthesis in "Past Query Outcomes," which affects the clarity and professionalism of the presentation.
>
> **A2**: Thank you again for catching this. **We have fixed the missing parenthesis issue in the updated draft.**
>
> > **Q3**: In the "Zero-shot Generalization" section, there is a misexpression where it states "solely on DOTA on xView".
>
> **A3**: Thank you, we apologize for the oversight and the incorrect phrasing. The "on xview" part was included by mistake, and **we have already fixed it in the updated version of the draft.**
>
> > **Q4**: The paper’s ablation study on the reward function lacks the combination of $R^{AS}+R^{LU}$, which could provide deeper insights into the impact of these components on model performance. Why was the combination of $R^{AS}+R^{LU}$ omitted in the reward function ablation study?
>
> **A4**: We fully agree with your point. Following your suggestion, we conducted an additional experiment where we trained DiffVAS using the $R^{AS} + R^{LU}$ reward function and compared its performance with other reward function variations. **These additional experimental results have already been included in Appendix Section A11 of the updated draft.**
>
> |Reward |Test with z={Ship}|Test with z={LV}|Test with z={Plane}
> |-----|--------|-------|-------|
> ||B =  5   7   10|B =  5   7   10|B =  5   7   10|
> |$R^{AS}$|1.65   2.71   3.77|1.89   2.85   3.90|2.05   3.50   4.68|
> |$R^{AS} + R^{LU}$|1.71   2.79   3.79|1.90   2.92   4.11|2.09   3.53   4.74|
>
> We observe a slight improvement in search performance when incorporating $R^{LU}$ into the reward function for training the TCPM module, underscoring the importance of the local uncertainty-based reward factor ($R^{LU}$). Finally, note that the full reward is still the best, as seen by comparing it with Table 5 in the main paper.
>
> > **Q5**: Is the presence of issues in writing and formatting, such as text descriptions and figure/table formats, an indication that the paper was prepared somewhat hastily?
>
> **A5**: Absolutely not! We sincerely apologize for the formatting and writing mistakes. Please rest assured that we will carefully address and resolve all such issues in the final version, to ensure a polished and well-presented paper.

---

> > ### Comment · Reviewer_uags · 2024-11-26
> > **Response to Rebuttal**
> >
> > I appreciate the authors' response. Most of my concerns have been resolved, and I will continue to maintain a positive score for the paper.

---

### Official Review · Reviewer_WaJg · 2024-11-02

**Soundness:** 2
**Presentation:** 2
**Contribution:** 2
**Rating:** 5
**Confidence:** 3

**Summary:**

Visual Active Search (VAS) is a framework for directing aerial exploration to pinpoint areas of interest using visual cues. Traditional VAS methods assume full knowledge of the search space and are tailored to specific target objects, limiting their practicality and versatility. This work introduces DiffVAS, a target-conditioned policy that searches for multiple object categories simultaneously in partially observable environments. DiffVAS uses a diffusion model to reconstruct the entire geospatial area from partial observations, allowing a reinforcement learning-based planning module to guide the search effectively.

**Strengths:**

+ Towards real-world situations, this work attempt to make decisions with incomplete information, the idea is  interesting and  practical.
+Experimental results are sufficient.

**Weaknesses:**

My main concern about this article lies in the mismatch between the motivation and experimental validation. The author emphasizes that the research is motivated by the challenge of obtaining complete information in the more realistic world. This phenomenon is indeed widespread, and the method based on diffusion models is also suitable for reconstructing more complete information from incomplete information, with a technically sound framework. However, my main concerns are those the current experimental data and environmental settings may not adequately simulate the incomplete data situations in real-world scenarios, and there is even a relatively large gap, which makes it difficult to directly apply the model trained in this article to real-world environments.

If the model cannot be directly applied to the real world, then the novelty of the technical framework itself seems not that strong. The idea of reconstructing the whole from parts has already been explored in many published works. The approach overlaps significantly with DiffMAE(ICCV23), which combines MAE with diffusion models for image reconstruction. The contribution seems incremental without clear novelty. The method closely resembles MDT(ICCV23), which uses unmasked tokens to predict masked ones while preserving diffusion training.

**Questions:**

My first question is about the practicality in the real world, if the author can convince me that this work can be applied in the real world, then I won't have any further questions.
Otherwise, I think that the current technical framework itself lacks sufficient novelty. I know we can find detailed differences between papers, but hope the author could clarify the core technological novelty of this paper.

---

> ### Author Response · Authors · 2024-11-14
>
> We appreciate the reviewer\'s comments and address all the concerns below.
>
> > **Q1**: The idea of reconstructing the whole from parts has already been explored in many published works. The approach overlaps significantly with DiffMAE(ICCV23), which combines MAE with diffusion models for image reconstruction. The contribution seems incremental without clear novelty. The method closely resembles MDT(ICCV23), which uses unmasked tokens to predict masked ones while preserving diffusion training.
>
> **A1**: **Fundamental difference with prior work**: The problem we are tackling in this work are fundamentally different than the mentioned work, such as MDT (ICCV23) and DiffMAE (ICCV23). Both these works aim to reconstruct an image from the partially observed scene, while we tackle a more challenging problem of active discovery of target objects. Hence, these methods (such as MDT, and DiffMAE) typically focus solely on optimizing for reconstruction, while our ultimate goal is identifying target-rich regions. Success for our task hinges on balancing exploration (obtaining useful information about the scene) and exploitation (finding objects of interest). To justify why such a reconstruction-only method is not appropriate to tackle active search problems, we performed an experiment with Greedy-DiffVAS that utilized the reconstructed features only to decide the next query location and compared the performance with our proposed framework. The empirical outcomes highlight the critical role of the planning module (TCPM) in learning an efficient search policy in partially observable environments. Please refer to lines 441- 460 in the manuscript for a discussion of this.
>
> > **Q2**: My first question is about the practicality in the real world, if the author can convince me that this work can be applied in the real world, then I won\'t have any further questions. Otherwise, I think that the current technical framework itself lacks sufficient novelty. I know we can find detailed differences between papers, but hope the author could clarify the core technological novelty of this paper.
>
> **A2**: **Real-world practicality:** It is not our view that novel research has to be able to run in a fully online, real-world setting -- that gap is quite unrealistic to be expected to be filled within a single paper that starts off at a new idea and establishes important foundations, in our opinion. Our method is at this research stage not yet implemented to run e.g. on a physical drone, and there are naturally quite a few steps to transition the code to run appropriately in such a real-world setup. Challenges to tackle include e.g. **(i)** transfer models trained in emulated satellite-data setting to use drone imagery; **(ii)** handle imperfect drone actuation and blurry imagery due to movements; **(iii)** ability to run on-device using limited compute. As for (iii), for example, one can leverage recent rapid advancements in diffusion models that make them more compute and time-efficient by reducing the number of reverse diffusion steps (such as [1,2] ). In principle, however, the approach could be proof-of-concepted even on actual drones, especially if one at the early stage allows for non-real-time processing.
>
> **Novelty of our work**: We would like to highlight the main contributions and novelties of our work:
>
> (1) We introduce the novel task of target-conditioned visual active search in partially observable environments. This is different to prior work which assumes full scene observability.
>
> (2) To address this task, we propose a novel framework that combines a diffusion-based Conditional Generative Module with an RL-based target-conditioned Planning Module to reason effectively in these partially observable environments. We design a novel reward function to guide the TCPM module in mastering an efficient search strategy that requires balancing exploration (query regions to gather important information about the environment) and exploitation (query regions based on the current belief to maximize target discovery) in partially observed scenes.
>
> (3) Additionally, our proposed inference strategy enables the DiffVAS framework to handle search tasks involving multiple target categories, which is different from prior work that focuses on single-target settings.
>
> Our primary focus in this work has been to lay the foundation for the novel and real-world relevant task of Target-Conditioned Visual Active Search in Partially Observable Environments, which we term TC-POVAS. We hope our work serves as a catalyst for future advancements in the field of "Active Search in Partially Observable Environments".
>
> **References:**
>
> 1. Frans, Kevin, et al. "One Step Diffusion via Shortcut Models." arXiv preprint arXiv:2410.12557 (2024).
>
> 2. Delbracio, Mauricio, and Peyman Milanfar. "Inversion by direct iteration: An alternative to denoising diffusion for image restoration." arXiv preprint arXiv:2303.11435 (2023).

---

> > ### Author Response · Authors · 2024-12-01
> > **Reminder to Reviewer WaJg**
> >
> > Dear Reviewer WaJg,
> >
> > As the discussion period approaches its conclusion, we would like to follow up to ensure that all your concerns have been fully addressed. If you require any further clarifications, please do not hesitate to let us know, and we will be glad to provide them promptly. Thank you for all your time and effort.
> >
> > Regards,
> >
> > Authors

---

### Official Review · Reviewer_yyn7 · 2024-11-03

**Soundness:** 2
**Presentation:** 3
**Contribution:** 3
**Rating:** 5
**Confidence:** 4

**Summary:**

This paper focuses on the visual active search (VAS) problem. Current work usually assumes that the global search space is known, which is not consistent with practical VAS applications. Accordingly, the authors formulate the target-conditioned Visual Active Search in Partially Observable environments problem, which points out that the agent needs to explore the environment step by step while searching for the Target of the specified category.

Based on the above problems, this paper proposes to use the diffusion model to generate a panoramic search space map based on the existing observations, and construct a target-conditioned planning module to select the appropriate region according to the generated image. This paper trains and tests on DOTA and other datasets, and compares with other methods. The results show that the proposed method can achieve good results in single-class/multi-class/zero-shot settings.

**Strengths:**

This paper has the following strengths:
1. Good originality. This paper starts from a very practical problem, that is, in the real environment, the UAV cannot know all the information of the search environment map in advance, so it can only make decisions within the existing knowledge. This is a good motivation for the question. Therefore, the article has good originality.
2. Simple and logically sound method. In view of the incompletely explored characteristics, this paper proposes CGM model, which uses diffusion model to generate complete scene information. Then, the target exploration strategy is trained, and the exploration effect and the scene information construction result are used as the reward function. This is a simple yet effective approach in active exploration.
3. Good clarity. This paper elaborates and analyzes the problem of self-definition in detail. And a large number of quantitative methods are used to describe the various details of the method, such as how to construct the reward function, the details of the loss function and so on. At the same time, the overall expression logic of the paper is relatively clear, and a large number of experiments prove the effectiveness of the method.

**Weaknesses:**

This paper has the following weaknesses:
1. Problem setting to be optimized.  This paper points out that the action space is a patch that can reach any patch in the search area. After executing the action, the visual information of the patch will be added to the prior knowledge of the UAV. However, in fact, the UAV can also obtain the visual information of the path position during the movement. Taking Figure 1 in the article as an example, from grid1 to grid5, the UAV should be able to obtain the visual information of grid2-4 at the same time, not only the visual information of grid5. Therefore, based on the actual scene and the rationality of the action space, the problem setting in this paper needs to be further optimized.
2. Details on training and testing are missing: For a series of experiments in this paper, the details of how the method is trained need to be further described. The lack of such descriptive information may prevent subsequent related work from being further developed above.
3. The motivation of the experimental setup needs further clarification.  The paper sets up single-class, multi-class and zero-shot experiments. However, the multiple-target category only gives results for three combinations, and zero-shot generalization cannot determine whether the category is the one that has appeared in the training phase. The experimental Settings need to be further supplemented.

**Questions:**

1. As far as I know, the methods you compare in this article, such as E2EVAS and MPS-VAS, are exploratory work with prior information about the scene. Please give the details of how to infer the above method under your problem setting, that is, without the full information of the scene.
2. Please provide motivation for the experimental setup, especially the class setup on multi-class experiments and zero-shot experiments.
3. Please provide more details about the training. In particular, how you've done policy training on datasets like DOTA. How do you sample episodes during training.

---

> ### Author Response · Authors · 2024-11-14
>
> Thanks for your detailed reviews and thoughtful questions. Below, we address all your comments.
>
> > **Q1** Problem setting to be optimized.
>
> **A1** There are certainly variations of our task setup that could be explored, each with its own trade-offs and assumptions (for example, a third setting would be one where the agent can only move to an adjacent position in each action). The task setup you suggest, where the agent gets to observe all patches between its previous and current patch, has the risk of leading to fairly uninteresting search behaviors, where a (close-to) optimal strategy may be to always move to a maximally distant uncovered location (since on average, that implies as many revealed subareas as possible, which in turn leads to on average more targets being discovered). Furthermore, when the agent makes a movement that is not vertical or horizontal, there are multiple possible paths that could be filled in, thus becoming a bit ill-defined how to do the fill-in in many cases. We therefore believe our current task formulation is reasonable, but we do recognize that parts of the suggestion could be incorporated in future work, although arguably the "intermediate" revealed sub-areas ought to be blurred due to movement.
>
> > **Q2** Details on training and testing are missing.
>
> **A2** We discuss the implementation details in Appendix section A4. Also, in the main paper starting on Line 383, we write: "Both xView and DOTA are satellite image datasets, with roughly 3000 px per dimension and representing approximately 60 object categories. We use 50%, 17%, and 33% of the large satellite images to train, validate, and test the methods, respectively". **For convenience, we provide the anonymous link to the script for training and inference of DiffVAS in Appendix section A4 of the updated draft.** We are committed to open-sourcing the GitHub link upon acceptance, including all model weights, to facilitate an easy reproduction of all our experimental results. We would be happy to provide any further details about our experiments if you require any additional specifics or clarification.
>
> > **Q3** Details about inference with E2EVAS and MPS-VAS.
>
> **A3** During inference, at each search step, we provide only the partially observed glimpses to the policy network, rather than the full search image. Specifically, we input an image with black patches representing unobserved areas and the ground truth content for the already observed regions. Based on the policy's action, we then update the input image by adding the newly revealed patch, which corresponds to the patch selected by the policy for querying. Aside from that, the E2EVAS and MPSVAS remain unchanged from the original version.
>
> > **Q4** Motivate the experimental setup in multi-target and zero-shot setting.
>
> **A4** In both xView and DOTA, **due to the large combinatorial space of possible category subsets, we selected only a representative subset for evaluation**. Moreover, certain classes are extremely rare within the dataset. For instance, images containing at least one instance of class Cement-Mixer appear only once. Consequently, including such classes as targets is not meaningful, as evaluating performance based on a single image does not provide robust analysis/results. Furthermore, regarding the multi-target results, in addition to those reported in Table 2 of the main paper, Table 9 in the Appendix includes 3 more multi-target combinations, bringing the total to 6 combinations in this work.
>
> The zero-shot results are presented starting around line 494 in the main paper. In this section, we train models on DOTA using categories that do not appear in xView. Then, in Table 7, we report the zero-shot performance on the unseen xView classes. **These additional details have been included in Appendix A9 of the updated draft.**
>
> > **Q5** How do you sample episodes during training?
>
> **A5** The details of our training and evaluation settings are provided in Appendix A4. During training, we randomly sample an image from the training set and select a random budget within the range {1 to N-1} (where N is the number of grids). We then extract the set of target categories if at least one instance of each category is present in the image, based on the ground truth annotations (which also include the precise locations of these target objects as a bounding box). Once the target category set is determined, we randomly select a category from this set to train our policy. The rationale for using a random-budget and random-target is to ensure that the policy learns to be robust against the target category and budget constraint. Across various experiments, we demonstrate that DiffVAS consistently outperforms the baseline across different budgets and target categories. **These additional details have already been included in Appendix A10 of the updated draft**. Moreover, we would be happy to incorporate any other specific training details you suggest for the final version.

---

> ### Author Response · Authors · 2024-12-01
> **Reminder to Reviewer yyn7**
>
> Dear Reviewer yyn7,
>
> As the discussion period approaches its conclusion, we would like to follow up to ensure that all your concerns have been fully addressed. If you require any further clarifications, please do not hesitate to let us know, and we will be glad to provide them promptly. Thank you for all your time and effort.
>
>
> Regards,
>
> Authors

---

### Author Response · Authors · 2024-11-14
**Global Response**

Dear Reviewers,

We thank all the reviewers for their valuable feedback, which has helped improve our paper. We are pleased that reviewers find our problem interesting and that no technical concerns were raised. We have carefully considered all reviewers' feedback and thoroughly updated the Main Paper and the Appendix, ensuring that all concerns are fully addressed. Thank you for your attention.

Thanks and Regards,

Authors

---

> ### Author Response · Authors · 2024-11-21
> **Global Response - 2**
>
> Dear Reviewers,
>
> Since we are now past the halfway point of the discussion period, we want to ensure that our responses have fully addressed your concerns. Please let us know if any further clarification or details are needed, and we would be happy to provide them as quickly as possible. If you believe our response has adequately addressed the concerns you raised, we kindly ask you to consider the possibility of raising the score. Thank you for your time.
>
>
> Regards,
>
> Authors

---

### Meta-Review · Area_Chair_DZsC · 2024-12-16

**Metareview:**

This paper addresses the Visual Active Search problem and received mixed ratings of 6, 5, 5, 5.
During the rebuttal phase, only reviewer uags engaged with the authors, and none of the reviewers participated in the AC-reviewer discussion.
The AC reviewed the authors' response, which addressed some concerns. However, key issues raised by the reviewers remain unresolved, including the problem setting to be optimized (reviewer yyn7, WaJg) and infeasibility of deployment in real-world scenarios (reviewers WaJg and BQ2P).

Given the highly competitive standards of ICLR, the AC recommends rejection of this submission. The authors are encouraged to incorporate the reviewers' feedback and consider resubmitting to a next venue.

**Additional Comments On Reviewer Discussion:**

During the rebuttal phase, only reviewer uags engaged with the authors, and none of the reviewers participated in the AC-reviewer discussion.
The AC reviewed the authors' response, which addressed some concerns. However, key issues raised by the reviewers remain unresolved, including the problem setting to be optimized (reviewer yyn7, WaJg) and infeasibility of deployment in real-world scenarios (reviewers WaJg and BQ2P).

---

### Decision · Program_Chairs · 2025-01-22

Reject